# UFO: A Unified Approach to Fine-grained Visual Perception via Open-ended Language Interface

**Hao Tang**[1,2]    **Chenwei Xie**[2]    **Haiyang Wang**[1]    **Xiaoyi Bao**[2,3]
**Tingyu Weng**[2]    **Pandeng Li**[2]    **Yun Zheng**[2†]    **Liwei Wang**[1,4,5†]
[1]Center for Data Science, Peking University  [2]Alibaba Group
[3] CASIA   [4] Center for Machine Learning Research, Peking University
[5] State Key Laboratory of General Artificial Intelligence, Peking University, Beijing, China
{tanghao@stu, wanghaiyang@stu, wanglw@cis}.pku.edu.cn   baoxiaoyi2021@ia.ac.cn
{eniac.xcw, wengtingyu.wty, lipandeng.lpd, zhengyun.zy}@alibaba-inc.com

## Abstract

Generalist models have achieved remarkable success in both language and vision-language tasks, showcasing the potential of unified modeling. However, effectively integrating fine-grained perception tasks like detection and segmentation into these models remains a significant challenge. This is primarily because these tasks often rely heavily on task-specific designs and architectures that can complicate the modeling process. To address this challenge, we present UFO, a framework that **U**nifies **F**ine-grained visual perception tasks through an **O**pen-ended language interface. By transforming all perception targets into the language space, UFO unifies object-level detection, pixel-level segmentation, and image-level vision-language tasks into a single model. Additionally, we introduce a novel embedding retrieval approach that relies solely on the language interface to support segmentation tasks. Our framework bridges the gap between fine-grained perception and vision-language tasks, significantly simplifying architectural design and training strategies while achieving comparable or superior performance to methods with intricate task-specific designs. After multi-task training on five standard visual perception datasets, UFO outperforms the previous state-of-the-art generalist models by 12.3 mAP on COCO instance segmentation and 3.3 mIoU on ADE20K semantic segmentation. Furthermore, our method seamlessly integrates with existing MLLMs, effectively combining fine-grained perception capabilities with their advanced language abilities, thereby enabling more challenging tasks such as reasoning segmentation. Code and models are available at https://github.com/nnnth/UFO.

## 1  Introduction

Multimodal large language models (MLLMs) [51, 92, 40, 2, 12, 46] have made significant progress, exhibiting outstanding performance on various visual tasks. Despite these achievements, their scopes are largely confined to image-level vision-language tasks, leaving fine-grained perception (*e.g.*, detection and segmentation) as a critical weakness. Recent studies have shown that enabling MLLMs to collaborate with off-the-shelf detectors and segmenters can enhance precise visual understanding [80, 18] and facilitate advanced applications such as mobile agents [71, 70, 82, 41], indicating that endowing MLLMs with fine-grained perception capabilities is beneficial. However, seamlessly integrating these tasks into MLLMs poses challenges because traditional specialized methods heavily rely on complex and task-specific designs, such as RPN [59] and mask decoders [34].

---

[†]Corresponding author.

39th Conference on Neural Information Processing Systems (NeurIPS 2025).

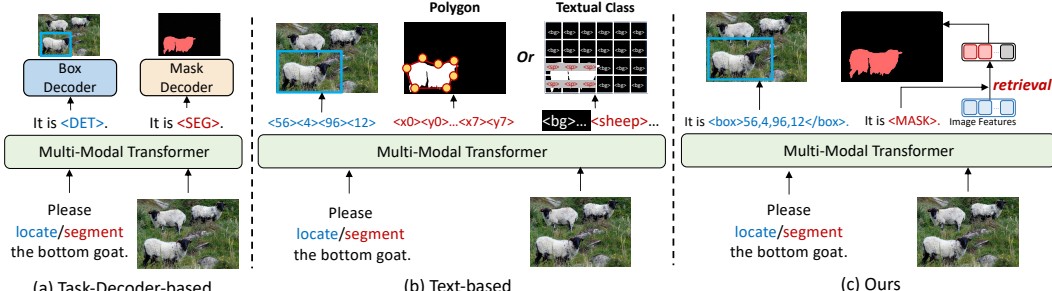

Figure 1: Methods to augment MLLMs with fine-grained perception tasks. (a) Relying on task decoders [37, 77], (b) Previous text-based methods represent boxes with location tokens [52] and represent masks with suboptimal polygons [74, 69] or textual classes [69, 38], (c) Ours: predicting open-ended text sequences while using a simple yet effective embedding retrieval approach for masks.

Many existing works [37, 77, 57, 60, 53, 89, 11] augment MLLMs with task-specific decoders, such as LISA [37] using SAM for segmentation or VisionLLM v2 [77] adding box and mask decoders. However, this combination introduces several limitations. First, task decoders add architectural complexity, necessitating compatibility among multiple components whenever the LLM is scaled up or new task decoders are introduced. Second, it complicates the end-to-end training. For example, the last stage in VisionLLM v2 [77] is dedicated to finetuning the task decoders because they fail to converge in earlier stages. These issues create a significant discrepancy with traditional vision-language modeling, limiting their broader application in general-purpose MLLMs. To remove task decoders, another line of research [52, 7, 81, 74, 54] converts boxes into location tokens or textual numbers and transform masks into polygon vertices. However, using a limited number of vertices for masks introduces quantization errors, especially for masks with complex shapes and multiple regions.

To address the above limitations, GiT [69] uses two mask representations: for instance segmentation, it uses polygons, and for semantic segmentation, it predicts textual classes for each pixel, as shown in Figure 1 (b). However, it still falls short of unifying fine-grained perception tasks into MLLMs. Firstly, although textual class can represent masks with any shape, it results in overly long sequences and slow inference. Secondly, to achieve better performance, GiT sets specific vocabularies for each task (e.g., detection can only output location tokens), which is incompatible with open-ended text generation. Finally, GiT does not scale to MLLMs and uses a Vision Transformer (ViT [21]) for multimodal tasks, resulting in poor language abilities. Hence, it is essential to develop a more effective approach to unify fine-grained perception into MLLMs. This method should effortlessly align with open-ended language interfaces, involve minimal structural complexity and deliver excellent performance.

In this paper, we present UFO, which unifies fine-grained perception tasks through the same open-ended language interface as vision-language tasks, without any task decoders. By carefully organizing and translating all task outputs into open-ended text sequences, we demonstrate that competitive performance can be achieved without complex task-specific designs. As illustrated in Figure 1 (c), we reformulate segmentation as an embedding retrieval problem, where the mask token embedding computes similarity with image features by dot product, retrieving high-similarity positions to generate the mask. This design effectively leverages the output image features processed by MLLMs, which are often overlooked in previous methods. Our intuition is that since MLLMs achieve strong visual understanding, the mask information is already in the image features and we just need to retrieve it. Furthermore, we introduce a novel method that upsamples output masks by predicting multiple mask tokens, resulting in more refined masks and improved performance. Thanks to this strategy, we can efficiently and accurately represent masks of any shape using only **16** tokens.

We first validate our method following GiT [69], which uses a lightweight ViT but can share the same formulation as MLLMs (see Table 1), allowing efficient validation. GiT constructs a comprehensive multi-task benchmark, which covers various granularity fine-grained perception tasks. Under the same evaluation protocols, UFO outperforms GiT by **12.3** mAP and **3.3** mIoU in COCO instance segmentation and ADE20K semantic segmentation (see Table 2). We then scale our method to MLLMs to integrate language abilities with fine-grained perception. As shown in Table 3, UFO achieves competitive results in visual grounding without decoders or polygon approximations. Benefiting from the shared representations of the open language interface, UFO can deeply unify textual reasoning and image segmentation, surpassing the previous state-of-the-art method on the challenging ReasonSeg [37] benchmarks by **6.2** gIoU (see Table 4).

In summary, our contributions are listed as follows:

(1) We introduce UFO, a unified framework for diverse fine-grained perception tasks through the same open-ended language interface as vision-language tasks, without task-specific decoders.

(2) We reformulate segmentation as an embedding retrieval problem, exploring both text generation and image representation abilities of the language interface, significantly outperforming previous text-based methods on instance and semantic segmentation tasks.

(3) Our framework seamlessly integrates with MLLMs, delivering better performance than previous state-of-the-art methods on the ReasonSeg benchmarks.

## 2 Related Work

### 2.1 Multimodal Large Language Models

Inspired by the success of large language models (LLMs), multimodal large language models (MLLMs) have rapidly advanced in recent years. Early efforts [43, 92, 17] finetune LLMs with instruction datasets, demonstrating strong multimodal understanding. More advanced MLLMs like Qwen2.5-VL [2], and InternVL2.5 [12] have emerged recently, offering superior multimodal comprehension through larger model sizes and extensive training data. However, these models mainly focus on image-level vision-language tasks, with less exploration of fine-grained visual perception.

### 2.2 Extend MLLMs with Fine-grained Perception

**Extend MLLMs with Task Decoders.** Recent works [37, 77, 57, 60, 3, 87, 53, 78, 88, 84, 89, 76] introduce task decoders to extend MLLMs with tasks like detection and segmentation. These models treat the MLLM as a coarse proposal extractor, passing the task-relevant embeddings to specialized decoders. The decoders then manage task-specific details, such as regressing boxes or generating masks. Although this approach yields strong performance, extra task decoders complicate architectures and training, undermining the unified design of MLLMs and limiting their potential. Recently, HiMTok [73] reformulates segmentation as mask image generation. However, this approach still requires training a specialized VQ decoder, increasing both training and structural complexity.

**Extend MLLMs with Text Outputs.** For object-level tasks, previous methods [52, 7, 81, 68, 74, 6, 54] have employed location tokens or textual numbers to represent boxes. For pixel-level tasks such as segmentation, a common format is polygonal approximation [74, 54]. Although VistaLLM [54] reduces the errors of polygons through adaptive sampling, it is inadequate for general segmentation tasks. First, polygons struggle to accurately represent "stuff" categories with amorphous regions (e.g., roads with parked cars), which are common in real world [90, 4, 16]. Second, polygons inherently cause information loss, especially for detailed structures like retinal vessels [64]. Text4Seg [38] directly predicts textual labels for image patches but still requires an additional refiner (e.g., SAM [34]) to achieve better performance. In contrast, UFO leverages the multimodal outputs of MLLMs to generate precise masks for any shape, offering greater expressiveness and improved performance.

### 2.3 Vision Generalist Models

Vision generalist models aim to establish a unified framework supporting various vision-centric tasks. Inspired by the seq2seq framework in NLP, previous generalist models [72, 69, 9, 47] transform visual tasks into sequence generation problems. Notably, GiT [69] unifies five core visual tasks by language interface, supporting box, mask, and text outputs. However, these models typically focus solely on visual tasks and lack the advanced language capabilities required for complex reasoning [37].

## 3 Methods

As our method is applicable to various multimodal architectures, we first present a unified architectural abstraction in Section 3.1. Then, in Sections 3.2 and 3.3, we explain how to integrate box and mask representations into the open-ended language interface. Finally, in Section 3.4, we describe our multi-task data template for joint training.

Table 1: We abstract current multimodal architectures into three components: (1) Image tokenizer, converting images into visual tokens; (2) Text tokenizer, outputting text tokens; (3) Multimodal transformer, jointly processing visual and text tokens. We construct three variants by this formulation.

| Model | Image Tokenizer | Text Tokenizer | Multimodal Transformer |
|---|---|---|---|
| LLaVA [43] | CLIP [56],MLP | Llama Tokenizer [67] | Vicuna [14] |
| EVE [20] | Patch embedding | Llama Tokenizer [67] | Vicuna [14] |
| GiT [69] | Patch embedding | Bert Tokenizer [19] | ViT [21] |
| UFO-ViT | Patch Embedding | Bert Tokenizer [19] | ViT [21] |
| UFO-LLaVA-1.5-7B | CLIP [56],MLP | Llama Tokenizer [67] | Vicuna 1.5 [14] |
| UFO-InternVL2.5-8B | InternViT [12],MLP | InternLM2.5 Tokenizer [86] | InternLM2.5-7B [86] |

## 3.1 Preliminary

Our goal is to unify fine-grained perception tasks into the open-ended language interface, thereby ensuring compatibility with any multimodal architecture that supports the same interface. We abstract existing multimodal architecture into three components based on the modalities they process: image tokenizer, text tokenizer and multimodal transformer, as shown in Table 1. For example, in LLaVA [43], the image tokenizer includes a vision encoder and MLP connector that extract visual features and map them into the LLM's input space, while the multimodal transformer corresponds to the LLM. This abstraction applies not only to MLLMs with various image tokenizers [43, 40, 20] but also to vision generalist models with similar architectures [69], significantly broadening the scope of our method. To avoid confusion, we will refer to MLLMs by default in the following sections.

## 3.2 Bounding Box Representation

To align with the open-ended language interface while avoiding the addition of extra location tokens, we directly translate boxes into textual numbers. Each box is represented by the coordinates of its top-left $(x_1, y_1)$ and bottom-right $(x_2, y_2)$ corners. The continuous values of these coordinates are discretized into integers within [0, range], enclosed by <box> and </box> tokens. If a class label is required, we simply prepend the textual class before the <box> token. For example, a box of a person can be represented as: person,<box>465,268,589,344</box>. This method converts boxes to open-ended sequences, effectively aligning with vision-language tasks.

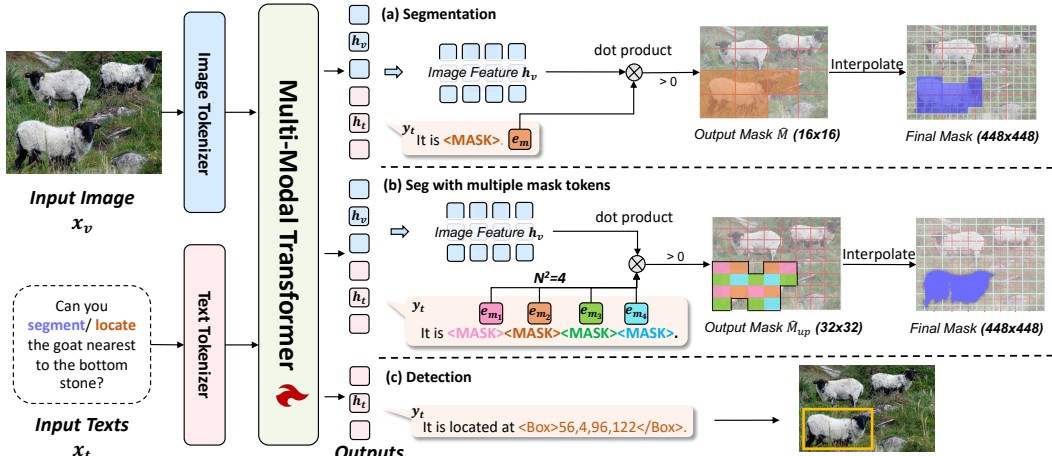

Figure 2: Overview of our approach. (a) Segmentation modeling: the mask token embedding retrieves similar image features to generate masks (shown with matching colors). (b) Upsampling masks by multiple mask tokens, retrieving more details by more tokens. We use $N$=2 to illustrate while using $N$=4 in implementation. (c) We output open-ended text sequences with textual numbers for detection.

## 3.3 Mask Representation

Representing masks via the language interface is more challenging because masks contain more detailed information than boxes. Previous methods either use polygon formats, which sacrifice details,

or assign textual classes to each pixel, resulting in overly long sequences. Therefore, a more efficient method to represent detailed masks is needed.

We observe that in MLLMs, the language interface is actually multimodal, where projected image features and text features are combined and jointly processed by the LLM. However, most existing methods ignore the output image features processed by the LLM. We argue that since MLLMs can express where and what objects are in text form, the mask information is already encoded in the image features. We just need to teach the model to decode this information. Therefore, we design a representation method based on image features and text embeddings. Instead of storing mask information in text embeddings, we use the text embeddings as query embeddings to extract mask information from the image features. The detailed approach is described below.

**Segmentation by Embedding Retrieval.** To incorporate the segmentation task using only the language interface, we reformulate it as an embedding retrieval problem. We first augment the basic vocabulary of the model with a `<MASK>` token, which serves as the indicator for mask generation. When performing segmentation, the model is trained to output the `<MASK>` token, as shown in Figure 2 (a). Formally, given an input image $\mathbf{x_v}$ and a segmentation prompt $\mathbf{x_t}$, the model $\mathcal{F}$ generates the text response $\mathbf{y_t}$ and corresponding output embeddings $\mathbf{h_t}$, image features $\mathbf{h_v}$ as:

$$\mathbf{h_v}, \mathbf{y_t}, \mathbf{h_t} = \mathcal{F}(\mathbf{x_v}, \mathbf{x_t}). \tag{1}$$

We extract the mask token embedding $\mathbf{e_m}$ corresponding to the `<MASK>` token from $\mathbf{h_t}$. To generate the segmentation mask, we compute the similarity between the mask token embedding $\mathbf{e_m}$ and the image features $\mathbf{h_v}$ via a scaled dot product. Positive scores are retrieved to form the binary mask $\hat{\mathbf{M}}$. This process is expressed as:

$$s = \frac{\mathbf{e_m} \mathbf{h_v}^\top}{\sqrt{d}}, \quad \hat{\mathbf{M}} = \mathbb{I}(s > 0), \tag{2}$$

where $d$ is the embedding dimension, $s$ represents the similarity scores, and $\mathbb{I}$ is the indicator function that converts the similarity scores into a binary mask. By computing the dot product similarity between the mask token embedding and image features, we retrieve the most relevant image features corresponding to the mask token, thereby producing a mask aligned with the original image.

Our approach leverages MLLMs' inherent capabilities for segmentation without task decoders. We hypothesize that, in well-encoded image features, features with the same semantics will group into clusters. Therefore, generating a mask token embedding equates to identifying the center of the relevant image feature cluster, while computing the similarity reflects this relationship. This approach can easily apply to other pixel-level tasks, such as depth estimation (see Table 20 in the appendix).

**Upsampling by Multiple Mask Tokens.** Due to the redundancy in visual information, it is common to process visual features at reduced resolutions. For example, the CLIP-L/14 [56] downsamples image features by a factor of 14. In above method, similarities are computed using downsampled image features, resulting in low-resolution masks. However, directly upsampling by interpolation leads to coarse results and suboptimal performance due to the high interpolation factor.

To address this issue, we propose an upsampling method by predicting multiple mask tokens. For an image $\mathbf{x_v} \in \mathbb{R}^{H \times W \times 3}$, we obtain image features $\mathbf{h_v} \in \mathbb{R}^{H_p \times W_p \times d}$ downsampled by the patch size $p$, where $d$ represents the feature dimension. Our target is to upsample the generated mask by $N$ times, producing $\hat{\mathbf{M}}_{\text{up}} \in \mathbb{R}^{(H_pN) \times (W_pN)}$ from image features $\mathbf{h_v} \in \mathbb{R}^{H_p \times W_p \times d}$. This requires decoding an $N \times N$ mask for each position in the image features. To achieve this, we train the model to autoregressively predict $N^2$ `<MASK>` tokens with embeddings $\{\mathbf{e_{m_i}}\}_{i=1}^{N^2}$. Each token corresponds to a single position in the $N \times N$ upsampling grid, as illustrated in Figure 2 (b). For each mask token embedding $\mathbf{e_{m_i}}$, we compute the similarity with the visual features $\mathbf{h_v}$:

$$s_i = \frac{\mathbf{e_{m_i}} \mathbf{h_v}^\top}{\sqrt{d}}, \tag{3}$$

where $\mathbf{e_{m_i}} \in \mathbb{R}^{1 \times d}$, $\mathbf{h_v}^\top \in \mathbb{R}^{d \times H_p \times W_p}$, and $s_i \in \mathbb{R}^{1 \times H_p \times W_p}$. These similarity scores $\{s_i\}_{i=1}^{N^2}$ are then concatenated and reshaped into an upsampled similarity map:

$$s_{\text{concat}} = \text{concat}(\{s_i\}_{i=1}^{N^2}), \quad s_{\text{concat}} \in \mathbb{R}^{N^2 \times H_p \times W_p}, \tag{4}$$

$$s_{\text{up}} = \text{reshape}(s_{\text{concat}}), \quad s_{\text{up}} \in \mathbb{R}^{(H_pN) \times (W_pN)}. \tag{5}$$

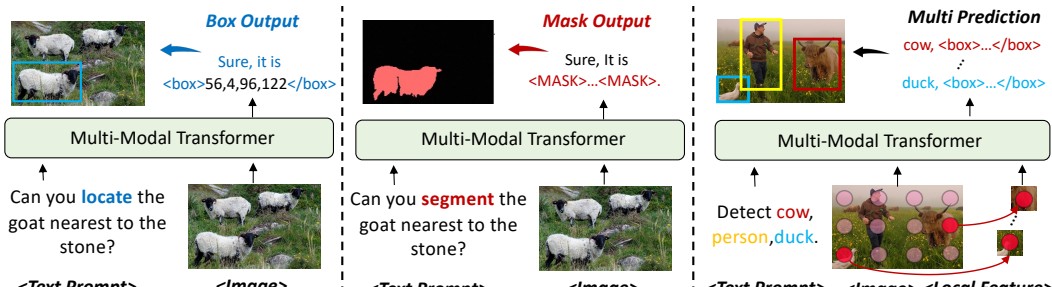

Figure 3: Multi-task data template examples. Red dots represent sampled grid point features, acting as local visual prompts for generating text sequences for nearby objects.

Finally, we retrieve positive scores in $s_{\text{up}}$ to generate the upsampled binary mask $\hat{\mathbf{M}}_{\text{up}}$. By default, we set $N = 4$, predicting 16 <MASK> tokens, which upsamples the output mask by a factor of 4. The mask is then aligned with the original image resolution through interpolation.

Our method effectively leverages mask token embeddings as upsampling parameters, offering greater flexibility than traditional methods like bilinear interpolation and transposed convolution. Bilinear interpolation uses non-learnable parameters, while transposed convolution allows for learnable parameters, the same parameters are applied to all images after training. In contrast, we use embeddings generated by the network as the parameters, which can be customized for each image. This approach enables the model to generate optimal upsampling parameters dynamically, enhancing flexibility while achieving better performance (see Table 6).

Note that our method is fully compatible with open-ended language interfaces. We refer "open-ended" as the capability to generate variable-length text sequences terminated by an end-of-sequence token. In our approach, while we require fixed-length <MASK> tokens for segmentation, the text generation itself remains open-ended. Only after the generation is complete, we check if the output contains <MASK> segments of the required length. Furthermore, the interaction with image features is also performed after text generation is finished.

### 3.4 Multi-Task Data Template

Based on the above designs, we construct multi-task data templates for joint training. We classify tasks into two categories based on prediction number: single-prediction tasks like grounding produce one box or mask, and multi-prediction tasks like object detection generate several boxes. Merging multiple outputs into a long sequence is inefficient and the order among them is hard to define, making autoregressive learning of the sequence difficult [8]. Therefore, we adopt a parallel decoding approach that splits multi-prediction tasks into independent subtasks, each handling one prediction in parallel. This strategy effectively accelerates inference and enhances task scalability.

**Single prediction.** For tasks only require a single prediction, our task template is: `<Text Prompt><Image><Text Response>`. As shown in Figure 3, we follow previous methods to construct text prompts and use our unified box and mask representation for text responses.

**Multiple predictions.** To efficiently support multi-prediction tasks, we split complex tasks into independent subtasks with single prediction, enabling parallel decoding within a batch. The key to achieving parallelism is to ensure all subtasks are independent. Typically, multiple boxes and masks correspond to different locations. Therefore, we introduce local image features in the input to differentiate these sub-tasks, serving as visual prompts. The template is structured as follows: `<Text Prompt><Image><Local><Text Response>`, where `<Local>` refers to local image features interpolated by grids sampled on the image. The core idea is that each grid point is responsible for detecting its spatially nearest objects. During training, each ground-truth object is assigned to its nearest grid point, while the remaining grid points are assigned to predict end-of-sequence tokens. During inference, as illustrated in Figure 3, we sample points over the image, typically of size $K \times K$, resulting in a total of $M = K^2$ points. We then interpolate the image features at each of the $M$ grid locations to extract distinct grid features. These grid features, along with the global image features and the text prompt, are fed into the LLM. The example input sequence is structured as follows:

$$\text{Detect cow, person, duck.} < \text{Image} >< \text{Local}_1 >< \text{Local}_2 > \ldots < \text{Local}_M > \qquad (6)$$

Table 2: Results on GiT [69]'s multi-task benchmark. "⋆" denotes the model is capable of the task but no number is reported. "-" means incapability. We highlight joint training improvements with **bold** font and follow [69] to list modules for specific functions.

| Methods | Specific Modules Examples | Num | #Params | Object Detection AP | AP$_{50}$ | AP$_{75}$ | Instance Seg AP | AP$_{50}$ | AP$_{75}$ | Semantic Seg mIoU(SS) | Captioning BLEU-4 | CIDEr | REC Acc@0.5 |
|---|---|---|---|---|---|---|---|---|---|---|---|---|---|
| *Specialist Models* | | | | | | | | | | | | | |
| Deformable-DETR [94] | RegressionHead | 5 | 40M | 45.4 | 64.7 | 49.0 | - | - | - | - | - | - | - |
| Mask R-CNN [26] | FPN,RPN | 6 | 46M | 41.0 | 61.7 | 44.9 | 37.1 | 58.4 | 40.1 | - | - | - | - |
| Polar Mask [79] | CenternessHead | 5 | 55M | - | - | - | 30.5 | 52.0 | 31.1 | - | - | - | - |
| Mask2Former [13] | PixelDecoder | 5 | 44M | - | - | - | 43.7 | - | - | 47.2 | - | - | - |
| VL-T5 [15] | Faster R-CNN | 3 | 440M | - | - | - | - | - | - | - | 34.5 | 116.5 | - |
| MDETR [30] | RoBERTa,DETR | 6 | 188M | - | - | - | - | - | - | - | - | - | 86.8 |
| *Generalist Models (MultiTask-Training)* | | | | | | | | | | | | | |
| Uni-Perceiver [95] | None | 1 | 124M | - | - | - | - | - | - | - | 32.0 | ⋆ | ⋆ |
| Uni-Perceiver-MoE [93] | None | 1 | 167M | - | - | - | - | - | - | - | 33.2 | ⋆ | ⋆ |
| VisionLLM-R50 [74] | Deform-DETR | 6 | 7B | 44.6 | 64.0 | 48.1 | 25.1 | 50.0 | 22.4 | - | 31.0 | 112.5 | 80.6 |
| GiT-B$_{single-task}$ [69] | None | 1 | 131M | 45.1 | 62.7 | 49.1 | 31.4 | 54.8 | 31.2 | 47.7 | 33.7 | 107.9 | 83.3 |
| GiT-B$_{multi-task}$ [69] | None | 1 | 131M | 46.7 | 64.2 | 50.7 | 31.9 | 56.4 | 31.4 | 47.8 | 35.4 | 112.6 | 85.8 |
| GiT-L$_{multi-task}$ [69] | None | 1 | 387M | 51.3 | 69.2 | 55.9 | 35.1 | 61.4 | 34.7 | 50.6 | 35.7 | 116.0 | 88.4 |
| GiT-H$_{multi-task}$ [69] | None | 1 | 756M | 52.9 | 71.0 | 57.8 | 35.8 | 62.6 | 35.6 | 52.4 | 36.2 | 118.2 | 89.2 |
| UFO-ViT-B$_{single-task}$ | None | 1 | 131M | 47.8 | 65.7 | 52.0 | 42.6 | 65.8 | 46.1 | 49.5 | 34.2 | 111.1 | 83.6 |
| UFO-ViT-B$_{multi-task}$ | None | 1 | 131M | 48.3 | 66.6 | 52.6 | 43.5 | 66.2 | 47.0 | 50.2 | 35.3 | 114.2 | 85.8 |
| *Improvement* (single→multi) | | | | **+0.5** | **+0.9** | **+0.6** | **+0.9** | **+0.4** | **+0.9** | **+0.7** | **+1.1** | **+3.1** | **+2.2** |
| UFO-ViT-L$_{multi-task}$ | None | 1 | 387M | 52.9 | 71.3 | 57.9 | 47.3 | 70.9 | 51.6 | 54.0 | 35.9 | 118.6 | 88.5 |
| UFO-ViT-H$_{multi-task}$ | None | 1 | 756M | **54.1** | **72.4** | **58.9** | **48.1** | **71.6** | **53.0** | **55.7** | 37.6 | 123.6 | 89.2 |
| UFO-InternVL2.5-8B$_{multi-task}$ | None | 1 | 8B | 52.3 | 71.7 | 56.5 | 45.8 | 69.5 | 49.7 | 54.6 | **39.6** | **131.6** | 90.4 |

To enforce the independence of predictions for each grid point, we modify the self-attention mask to isolate each grid feature from the others. This ensures that the generation for one point does not influence another. Then we start generating in an autoregressive manner, with the difference that we predict M tokens at each forward step instead of just one. The generation process looks like this:

$$< \text{Local}_1 > \ldots < \text{Local}_M > | < T_1^1 > \ldots < T_M^1 > | < T_1^2 > \ldots < T_M^2 > | \ldots \qquad (7)$$

where $T_i^j$ denotes the $j$-th generated token for the $i$-th grid sequence, and | distinguishes the tokens produced in each forward pass. This decoding strategy shares the same philosophy as blockwise prediction [65] in LLMs, accelerating inference by generating multiple tokens simultaneously.

After decoding, we obtain $M$ output sequences. For detection, some might identify objects (`Duck, <box>...` or `Cow, <box>...`), while others corresponding to empty regions will predict end-of-sequence tokens. For instance segmentation, the process is identical, with the textual box representations (`<box>...`) being replaced by mask tokens (`<MASK>...`). This approach not only enhances efficiency but also simplifies the problem by breaking it down into simple, localized prediction tasks.

# 4 Training

To ensure efficient validation and fair comparison, we first follow GiT [69], using a smaller ViT as the multimodal transformer for multi-task training across five standard visual perception tasks. We then scale to MLLMs, validating on the same multi-task benchmark. Finally, we enrich the data by incorporating more diverse datasets, enabling fine-grained instruction tuning of MLLMs. After instruction tuning, the fine-grained perception capabilities are seamlessly integrated with the robust language abilities of MLLMs, thereby applying to perception tasks that require advanced language capabilities, such as reasoning segmentation.

## 4.1 Multi-Task Training

**Architecture.** To ensure fair comparison and validate our effectiveness across various architectures, we conduct multi-task training using two variants: UFO-ViT and UFO-InternVL2.5-8B. UFO-ViT strictly follows GiT [69], employing a SAM [34]-pretrained ViT [21] and a text tokenizer from BERT [19]. It is available in three sizes: ViT-B, ViT-L, and ViT-H. UFO-InternVL2.5-8B utilizes the pretraining weight of InternVL2.5-8B [12], with detailed model specifications provided in Table 1.
**Datasets.** We use the same multi-task dataset as GiT: COCO 2017 [42] for object detection and instance segmentation, COCO Caption [10] for image captioning, the RefCOCO series [48, 83] for referring expression comprehension (REC), and ADE20K [90] for semantic segmentation.

Table 3: Comparison of referring expression comprehension (REC) and segmentation (RES) performance. Results on REC are reported based on P@0.5. Results for RES are reported based on cumulative IoU (cIoU). * denotes the model is specifically finetuned on the dataset.

| Methods | Referring Expression Comprehension (REC) | | | | | | | | | Referring Expression Segmentation (RES) | | | | | | | | |
|---|---|---|---|---|---|---|---|---|---|---|---|---|---|---|---|---|---|---|
| | RefCOCO | | | RefCOCO+ | | | RefCOCOg | | Avg | RefCOCO | | | RefCOCO+ | | | RefCOCOg | | Avg |
| | val | testA | testB | val | testA | testB | val | test | | val | testA | testB | val | testA | testB | val | test | |
| *MLLMs with Task Decoders* | | | | | | | | | | | | | | | | | | |
| GLaMM-7B [57] | - | - | - | - | - | - | - | - | - | 79.5 | **83.2** | 76.9 | 72.6 | 78.7 | 64.6 | 74.2 | 74.9 | 75.6 |
| SAM4MLLM-8B [11] | - | - | - | - | - | - | - | - | - | 79.8 | 82.7 | 74.7 | 74.6 | 80.0 | 67.2 | 75.5 | 76.4 | 76.4 |
| HiMTok-8B [73] | - | - | - | - | - | - | - | - | - | **81.1** | 81.2 | **79.2** | **77.1** | 78.8 | 71.5 | 75.8 | 76.7 | 77.7 |
| PerceptionGPT-7B [53] | 88.6 | 92.5 | 84.6 | 82.1 | 88.6 | 74.2 | 84.1 | 85.2 | 85.0 | 75.1 | 78.6 | 71.7 | 68.5 | 73.9 | 61.3 | 70.3 | 71.7 | 71.4 |
| VisionLLM v2 [77] | 90.0 | 93.1 | 87.1 | 81.1 | 87.3 | 74.5 | 85.0 | 86.4 | 85.6 | 79.2 | 82.3 | 77.0 | 68.9 | 75.8 | 61.8 | 73.3 | 74.8 | 74.1 |
| *MLLMs w/o Task Decoders* | | | | | | | | | | | | | | | | | | |
| Shirka-7B [7] | 87.0 | 90.6 | 80.2 | 81.6 | 87.4 | 72.1 | 82.3 | 82.2 | 82.9 | - | - | - | - | - | - | - | - | - |
| MiniGPT-v2-7B [6] | 88.1 | 91.3 | 84.3 | 79.6 | 85.5 | 73.3 | 84.2 | 84.3 | 83.8 | - | - | - | - | - | - | - | - | - |
| Ferret-v2-7B [85] | 92.8 | 94.7 | 88.7 | 87.4 | **92.8** | 79.3 | **89.4** | **89.3** | 89.3 | - | - | - | - | - | - | - | - | - |
| VistaLLM-7B [54] | 88.1 | 91.5 | 83.0 | 82.9 | 89.8 | 74.8 | 83.6 | 84.4 | 84.8 | 74.5 | 76.0 | 72.7 | 69.1 | 73.7 | 64.0 | 69.0 | 70.9 | 71.2 |
| UFO-LLaVA-1.5-7B | 90.2 | 93.5 | 87.3 | 84.4 | 90.3 | 78.7 | 86.4 | 86.8 | 87.2 | 77.2 | 80.1 | 76.4 | 71.8 | 77.9 | 70.2 | 74.1 | 73.5 | 75.2 |
| UFO-LLaVA-1.5-7B* | 91.1 | 93.7 | 88.6 | 85.5 | 90.5 | 79.9 | 87.3 | 87.2 | 88.0 | 77.9 | 81.1 | 77.0 | 72.5 | 78.5 | 71.4 | 75.6 | 74.1 | 76.0 |
| UFO-InternVL2.5-8B | 91.8 | 94.3 | 87.5 | 86.9 | 91.3 | 80.6 | 87.9 | 88.6 | 88.6 | 80.0 | 81.6 | 78.1 | 76.7 | 79.9 | 72.3 | 75.5 | 76.3 | 77.6 |
| UFO-InternVL2.5-8B* | **93.1** | **94.8** | **89.2** | **87.7** | 92.1 | **82.3** | 88.2 | 89.2 | **89.6** | 81.0 | 82.6 | 78.6 | **77.1** | **80.4** | **72.6** | **76.7** | **77.3** | **78.3** |

Table 4: Results (gIoU) on ReasonSeg test set. * using reasoning segmentation data in training.

| Methods | ReasonSeg | | |
|---|---|---|---|
| | overall | short query | long query |
| X-Decoder [96] | 21.7 | 20.4 | 22.2 |
| SEEM [97] | 24.3 | 20.1 | 25.6 |
| LISA-7B [37] | 36.8 | 37.6 | 36.6 |
| LISA-7B [37]* | 47.3 | 40.6 | 49.4 |
| Cores-7B [3] | 48.7 | 41.0 | 50.9 |
| Cores-7B [3]* | 52.4 | 44.2 | 55.0 |
| HiMTok-8B [73]* | 60.8 | - | - |
| UFO-LLaVA-1.5-7B | 54.4 | 41.2 | 58.5 |
| UFO-LLaVA-1.5-7B* | 58.8 | 46.5 | 62.7 |
| UFO-InternVL2.5-8B | 60.0 | 48.7 | 63.6 |
| UFO-InternVL2.5-8B* | **67.0** | **56.2** | **70.4** |

Table 5: Results on vision-language benchmarks.

| Models | GQA | MMBench | MMVP | HallBench |
|---|---|---|---|---|
| InternVL2.5-8B | 60.6 | **84.6** | **76.3** | 50.1 |
| UFO-InternVL2.5-8B | **60.8** | 84.2 | **76.3** | **50.7** |

Table 6: Mask token number ablation on UFO-ViT-B$_{single-task}$ for instance segmentation.

| $N^2$ | 1 | 4 | 16 | 25 |
|---|---|---|---|---|
| mAP | 38.9 | 41.3 | 42.6 | 42.9 |
| FPS | 7.0 | 5.9 | 3.6 | 2.8 |

Table 7: Ablation of open-ended decoding on UFO-ViT-B$_{single-task}$ for detection.

| Decoding Rule | Beam Search | Positive Predictions | Detection mAP |
|---|---|---|---|
| ✓ | | 9.1 | 43.0 |
| ✓ | | 100 | 45.1 |
| | ✓ | 67.0 | 45.1 |

## 4.2 Fine-grained Instruction Tuning

**Architecture.** To demonstrate that our method is applicable to various MLLMs, we use not only InternVL2.5-8B [12] but also the LLaVA-1.5-7B [44] for pretraining, specifically UFO-LLaVA-1.5-7B. Architecture details are in Table 1.

**Datasets.** To enhance the model's versatility, we enrich the training data to 2.5M across 6 tasks, including VQA data from [39], COCO-Stuff [4], LVIS [25], etc. We additionally add RES task on the basis of five tasks in multi-task training. More details of data composition are in Table 8.

# 5 Experiments

## 5.1 Experimental Settings

**Multi-Task Training Details.** To facilitate comparison with specialist models, we also conduct single-task training independently on five selected tasks. For both single-task and multi-task training, we use a batch size of 24 and employ the AdamW [33] optimizer with a cosine annealing schedule, setting the initial learning rate to 0.0002. More details are in the appendix.

**Fine-grained Instruction Tuning Details.** In training, we use a batch size of 32 with gradient accumulation set to 16, running on 8 NVIDIA A100 GPUs for 120K iterations. The AdamW [33] optimizer and a cosine annealing schedule are employed, with a learning rate of 0.0002 and weight decay of 0.01. For efficient training, we employ LoRA [27] with a rank of 8, freezing the image tokenizer while keeping only the LLM trainable. More details are in Appendix Table 12.

**Training Objectives.** All tasks utilize a CrossEntropy Loss as they are unified under the open-ended language interface. For segmentation tasks, we additionally apply focal loss [61] and dice loss to supervise the mask output. The final loss for segmentation tasks is expressed as:

$$\mathcal{L}_{seg} = \lambda_{CE}\mathcal{L}_{CE} + \lambda_{focal}\mathcal{L}_{focal} + \lambda_{dice}\mathcal{L}_{dice}$$

We find that setting all weights to 1 offers better overall performance. See appendix for more details.

## 5.2 Multi-Task Evaluation

We evaluate performance in both single-task and multi-task settings across five vision-centric tasks, benchmarking it against specialized and generalist models. Without task decoders, our model adapts to various tasks by the open-ended language interface and achieves outstanding performance.

**Comparison with Specialist Models.** As shown in Table 2, our single-task model effectively bridges the performance gap with specialized models, achieving superior performance. For example, we achieve 47.8 mAP in detection compared to 45.4 mAP with Deformable-DETR [94] and 49.5 mIoU in semantic segmentation against 47.2 mIoU with Mask2Former [13]. In instance segmentation, we also outperform specialized methods like Mask R-CNN [26] while matching Mask2Former [13].

**Comparison with Generalist Models.** To facilitate comparison with GiT [69], we adopt its one-stage training without task-specific tuning. This involves jointly training on a mixed dataset of the five tasks and directly testing on their respective validation or test sets. Table 2 shows that our model outperforms the previous leading generalist model, GiT, across all tasks, with the same pretraining and data. Notably, in the largest ViT size, we outperform GiT by 12.3 mAP on COCO instance segmentation and 3.3 mIoU on ADE20K semantic segmentation, demonstrating the superiority of our segmentation modeling. We also surpass GiT 5.3 CIDEr in captioning, primarily due to our shared vocabulary across all tasks, while GiT uses task-specific vocabularies, hindering the task synergy.

We also observe a multi-task synergy effect like GiT, with performance on instance segmentation improved by 0.9 mAP and captioning increased by 3.1 CIDEr. Our multi-task improvements on segmentation also outperform GiT (0.7 vs. 0.1 mIoU). We attribute this to unified modeling across segmentation tasks, whereas GiT employs separate methods for instance and semantic segmentation.

After scaling to MLLMs, we observe improved performance on captioning and REC, while other tasks remain comparable to the UFO-ViT-L. We speculate that this performance difference primarily arises from different pretraining. For UFO-ViT, we use SAM [34] pretraining, making it more aligned with detection and segmentation. In contrast, InternVL2.5-8B is mainly pre-trained on image-level vision-language tasks, which better suit captioning and REC.

## 5.3 Fine-grained Instruction Tuning Results

**Visual Grounding** can be categorized into referring expression comprehension (REC) and segmentation (RES). We comprehensively list the results for the two tasks in Table 3. We report results in two settings: direct evaluation after joint training and specifically finetuning. Without using box decoders, our best model can surpass the VisionLLM v2 [77] by an average of 3.0%. After specific finetuning, our model achieves comparable performance with the state-of-the-art method Ferret-v2-7B [85]. While all previous approaches rely on mask decoders or polygon approximations for segmentation, our method delivers superior or comparable performance without them. For instance, our InternVL2.5 variant outperforms the SAM4MLLM [11] by an average of 1.9 cIoU and matches with HiMTok [73]. These outcomes validate the effectiveness of our method, demonstrating that with proper task modeling, MLLMs can handle fine-grained perception tasks without task decoders.

**Reasoning Segmentation** (ReasonSeg) is a challenging benchmark introduced by LISA [37], which presents more sophisticated and nuanced instructions, requiring models to leverage world knowledge and engage in deeper logical reasoning. We report both zero-shot and finetuned results. As shown in Table 4, with the same pretraining, our InternVL2.5 variant outperforms HiMTok [73] by 6.2 gIoU in finetuned settings. Notably, both Cores [3] and HiMTok [73] design a CoT strategy to generate segmentation masks progressively: answer the question with text first and then perform segmentation on the answered objects. In contrast, our method achieves better performance without this strategy, which implies that we can effectively perform reasoning while generating mask embeddings.

Since ReasonSeg requires both reasoning and precise segmentation, we attribute our improvement to better task integration through unified modeling. In decoder-based methods, the MLLM handles only language reasoning and generates coarse segmentation prompts, relying on an additional mask decoder for finer segmentation. This leads to information loss and insufficient synergy. In our unified modeling, the MLLM manages both language reasoning and precise segmentation, allowing different task capabilities to fully integrate within a shared parameter space, thereby enhancing synergy.

**Visual Question Answering.** Table 5 presents the performance on four VQA benchmarks (GQA [28], MMBench [45], MMVP [66], HallusionBench [24]). Thanks to our unification with the language

interface, the model's original performance is essentially maintained. Notably, we achieved a 0.6 improvement on [24], which may indicate that fine-grained fine-tuning helps reduce hallucinations.

### 5.4 Ablation Study

**Number of Mask tokens.** We ablate the number of mask tokens on the COCO instance segmentation task. As shown in Table 6, using multiple tokens significantly improves performance compared to a single token, but the gains plateau after 16. Considering the increased training and inference costs, we set $N^2 = 16$ by default for balance. The visualization of mask tokens is in Appendix Figure 8.

**Open-ended decoding.** We explore the impact of our open-ended decoding on single-task detection. As noted in Section 3.4, we split object detection into sub-tasks for each grid point (e.g., 625 grid points for a 1120×1120 image), which leads to an imbalance between positive and negative samples due to more grid points than objects. Our method utilizes a standard vocabulary (e.g., BERT's 30,524 tokens) for open-ended decoding. This output space is much bigger than the range of positive classes (e.g., 80 in COCO), worsening class imbalance and reducing positive predictions in inference. As shown in Table 7, while using decoding rules like removing negative classes from the vocabulary [69] could force all outputs to be positive, this compromises our generality. Therefore, we use beam search [58], which allows the model to explore multiple potential sequences. This approach effectively increases positive predictions and improves performance. By default, we only apply beam search for COCO detection, instance segmentation, and image captioning.

## 6 Conclusion

In this paper, we present UFO, a unified approach for various fine-grained visual perception tasks with an open-ended language interface. We translate all perception targets into open-ended text sequences and introduce a novel embedding retrieval method for segmentation. Experiments show that our method can achieve excellent performance on MLLMs without requiring architecture modifications. Our unification fully aligns with vision-language tasks, providing a flexible, effective, and scalable solution to enhance the fine-grained perception capabilities of MLLMs, paving the way to build stronger and more general multimodal models.

## Acknowledgments

LW is supported by National Science and Technology Major Project (2022ZD0114902) and National Science Foundation of China (NSFC92470123, NSFC62276005).

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

## A   Technical Appendices

Our technical appendices provides detailed information, including implementation details (§B), dataset descriptions (§C), and training specifics (§D). Inference speed and additional ablation studies are covered in §E and §F, respectively. Experiments on more tasks and discussions with previous work are included in §G and §H. Qualitative results from two training setups and visualizations of multiple mask tokens are presented in §I. Broader impacts are discussed in §J.

## B   Implementation Details

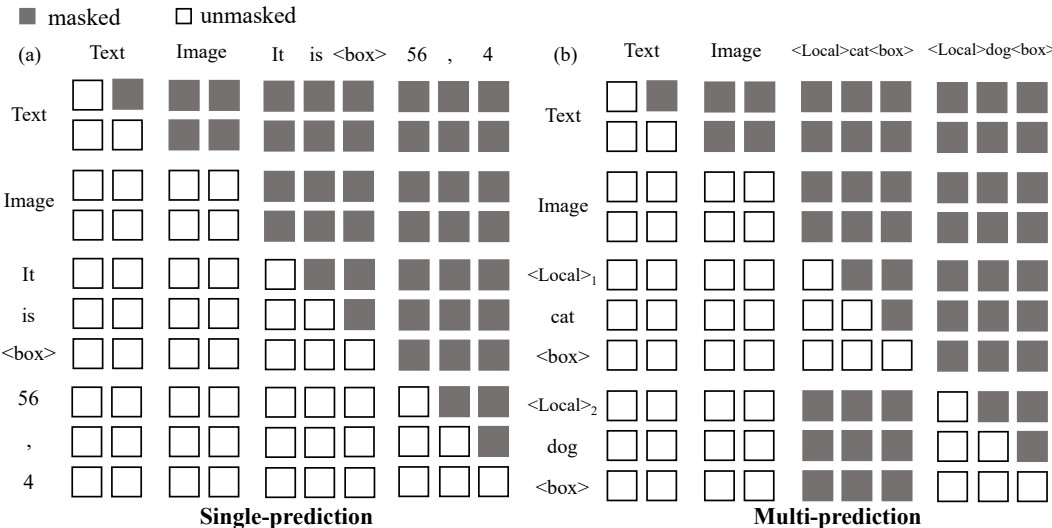

Figure 4: Attention mask visualizations. (a) We apply bidirectional attention for image features. (b) For multi-prediction tasks, we mask each subsequence from seeing others.

**ViT Architecture.** Our ViT architecture follows the same design as GiT [69], using the SAM variant of ViT. We also follow GiT to add 6 newly initialized transformer layers upon the original ViT, leading to better performance. For example, UFO-ViT-B consists of 18 layers.

**Grid Generation.** As mentioned in the multi-task template (Section 3.4), we divide multi-prediction tasks into sub-tasks, each corresponding to the nearest grid point. The number of grid points is roughly proportional to the image resolution, as shown in Table 9. Calculating the loss for all sub-tasks is computationally expensive. Therefore, we sample grid points and compute the loss on a subset, as presented in Table 9. When sampling, we prioritize positive samples. If the number of positive samples is less than the target size, we then sample from the negative samples. During training on MLLMs, we adjust the resolution based on the default resolution of the image tokenizer ($448^2$ for InternVL2.5 and $336^2$ for LLaVA-1.5) and modify the grid point configurations accordingly. For the default resolution of the two MLLMs, we use 100 grids and sample 40 of them.

**Attention Mask.** To accurately model multi-modal features and manage relationships among sub-tasks in multi-prediction tasks, we customize the attention mask based on autoregressive attention. As shown in Figure 4 (a), for tasks with a single prediction, we use autoregressive attention but apply bidirectional attention to the image features to better capture inter-image relationships. For tasks with multiple predictions, during training, we concatenate multiple sub-task sequences into a long sequence but use attention masks to prevent different sub-tasks from seeing each other, as illustrated in Figure 4 (b). This approach enables different sub-tasks to be decoded in parallel during inference. Assuming there are $M$ sub-tasks, we first process the <Text Prompt> and <Image> and save their key-value (KV) caches. These KV caches are then duplicated $M$ times to create caches for $M$ subsequences. By batching these subsequences together, the model can decode them in parallel, thereby accelerating the inference speed.

**Label Assignment.** In multi-prediction tasks, we use the Hungarian algorithm [35] to match sub-tasks with grid points, specifically to associate boxes and masks with grid points. For boxes, the

matching is based on the distance between the center of each box and the grid points. For masks, we first convert them into box format before matching.

**Post Processings.** Post-processing involves three steps. First, class names and box coordinates are extracted from the text sequence via pattern matching. Second, the confidence score is derived from the softmax probability of the first token predicted. For example, for the sequence "Duck, <box>...", we use the score associated with the "Duck" token. Finally, NMS is applied to filter the predictions.

**Coordinate Discretization.** We follow GiT [69] to convert continuous coordinates into discrete numbers within the range `[0, range]`. Specifically, we define the range as twice the image resolution. For example, for a $448 \times 448$ image, we convert coordinate values into integers within [0, 896].

**Data Augmentation.** For object detection, instance and semantic segmentation, we use `RandomFlip` and `RandomResizedCrop`. We also use `CopyPaste` for object detection and instance segmentation. For other tasks, we simply resize the images to the required resolution.

**Indicator.** In segmentation, we use sigmoid for indicator and then get a mask with a 0.5 threshold.

## C  Dataset Details

**Fine-grained Instruction Tuning Datasets.** The details of the datasets used for fine-grained fine-tuning are presented in Table 8. For LLaVA-1.5, we utilize the VQA data from the original paper [43], and for InternVL2.5, we employ the VQA data from LLaVA-OneVision [39]. Due to limited training iterations, we partially sampled some datasets, such as Objects365 and OpenImages, resulting in a final training dataset size of approximately 2.5M.

## D  Training Details

**Multi-Task Training Details.** Multi-task training setting is in Table 12. For single-task training, we simply reduce the training iterations to 120k. For multi-task training on InternVL2.5-8B, we keep all parameters trainable and reduce the iterations to 400k because of its faster convergence.

**Progressive High-resolution Training of MLLMs.** In multi-task training, object detection, instance segmentation, and semantic segmentation require predicting a large number of targets, including many small objects, making performance highly sensitive to resolution. For the relatively small UFO-ViT, we train directly using high resolution. However, for multi-task training on MLLMs, high resolution significantly increases training costs. Therefore, we adopt a progressive high-resolution training strategy: first training at a resolution of $448^2$ for 300k iterations, then at $896^2$ for 60k iterations, and finally at $1344^2$ for 40k iterations. We utilize InternVL2.5-8B's dynamic resolution to support high-resolution inputs. As shown in Table 10, increasing the resolution leads to substantial improvements in detection and segmentation performance, even with fewer iterations.

**Fine-grained Instruction Tuning Details.** Training settings are in Table 12. The training data for the MLLM includes six tasks, each containing multiple datasets. We use a sampling probability of 1/6 for REC, RES, Detection, and Instance Segmentation. For VQA and semantic segmentation, we apply sampling probabilities of 1/4 and 1/12, respectively, based on their data volumes. Within each task, sampling is conducted according to the size of the dataset. Additionally, inspired by high-resolution training in multi-task training, we first train with a low resolution (e.g., $448^2$ for InternVL2.5) for 90k iterations, and then switch to a high resolution (e.g., $896^2$ for InternVL2.5) for 30k iterations. When fine-tuning on a specific dataset, we maintain the same training setup but train for only 20k iterations.

**LoRA Configurations.** We use LoRA in fine-grained instruction tuning. Our trainable parameters include both the LoRA layers and text embeddings. As shown in the Table 13, UFO's LoRA parameter count is comparable to other models and much smaller than that of SAM4MLLM [11]. Unlike other methods, which also require training an extra mask decoder or even the entire LLM (e.g., HiMTok-8B [73], VisionLLMv2 [77], VistaLLM [54]), UFO achieves superior performance with fewer parameters. This highlights our parameter efficiency.

Table 8: Fine-grained instruction tuning datasets.

| Task | Sources | Size |
|---|---|---|
| VQA | *LLaVA-v1.5-mix665k [43] or LLaVA-OneVision [39](1M)* | 0.7M 1M |
| REC & RES | *RefCOCO [83], RefCOCO+ [83], RefCOCOg [83], RefCLEF [31]* | 85K |
| Object Detection | *Objects365 [62](200K), COCO [42], LVIS [25], nuImages [5]* | 0.6M |
| Instance Seg | *OpenImages [36](200K), LVIS [25], COCO [42],nuImages [5],* | 0.6M |
| Semantic Seg | *COCOStuff [4],Mapillary [49] nuImages [5], ADE20K [90]* | 0.3M |

Table 9: Resolution, grid number and sample grid number for the five tasks in multi-task training on UFO-ViT. Speed is measured on UFO-ViT-B, single A100 with batch size 1.

| Task | Resolution | Grid | Sample Grid | Speed |
|---|---|---|---|---|
| Object Detection | $1120^2$ | 625 | 250 | 4.1 |
| Instance Seg | $1120^2$ | 625 | 250 | 3.6 |
| Semantic Seg | $672^2$ | 225 | 90 | 4.8 |
| Image Captioning | $224^2$ | 0 | 0 | 7.7 |
| REC | $224^2$ | 0 | 0 | 9.1 |

Table 10: Performance of progressive high-resolution training on UFO-InternVL2.5-8B.

| Resolution | Iters | Detection | Ins Seg | Sem Seg |
|---|---|---|---|---|
| $448^2$ | 300k | 44.3 | 37.4 | 53.9 |
| $896^2$ | 60k | 51.7 | 44.1 | 54.6 |
| $1344^2$ | 40k | 52.3 | 45.8 | - |

Table 11: Inference speed on MLLMs. Speed is measured on an A100 GPU with batch size 1.

| Task | UFO-InternVL2.5-8B | UFO-LLaVA-1.5-7B |
|---|---|---|
| REC | 1.10 | 0.78 |
| RES | 0.58 | 0.67 |
| ReasonSeg | 0.57 | 0.65 |

Table 12: Multi-task training and instruction tuning settings.

| config | Multi-task (ViT) | Multi-task (MLLM) | Instruction tuning |
|---|---|---|---|
| optimizer | AdamW | AdamW | AdamW |
| learning rate | 2e-4 | 2e-4 | 2e-4 |
| weight decay | 0.05 | 0.01 | 0.01 |
| layer-wise lr decay | 0.85 | 0.85 | - |
| schedule | cosine | cosine | cosine |
| gradient norm clip | 0.1 | 1.0 | 1.0 |
| warmup iters | 1k | 1k | 1k |
| training iters | 640k | 400k | 90k+30k |
| batch size | 24 | 24 | 32 |
| gradient accumulation | - | - | 16 |
| LoRA rank | - | - | 8 |
| LoRA alpha | - | - | 16 |
| LoRA dropout | - | - | 0.05 |
| LoRA modules | - | - | LLMs |
| drop path | 0.1(B), 0.4(L,H) | - | - |
| precision | FP16 | BF16 | BF16 |
| GPUS | 24 × V100 | 8× A100 | 8× A100 |

Table 13: Comparison of trainable parameters when applying LoRA.

| Method | Base (M)LLM | LoRA Rank | LoRA Parameters | Text Embedding |
|---|---|---|---|---|
| Cores-7B [3] | LLaVA-7B [43] | 8 | 20M | 262M |
| GLaMM-7B [57] | Vicuna-7B [14] | 8 | 20M | 262M |
| SAM4MLLM-8B [11] | Qwen-VL-7B [1] | 256 | 693M | 1.24B |
| UFO-LLaVA-1.5-7B | LLaVA-1.5-7B [43] | 8 | 20M | 262M |
| UFO-InternVL2.5-8B | InternVL2.5-8B [12] | 8 | 19M | 758M |

Table 14: Ablation of beam search number on UFO-ViT-B$_{single-task}$.

| Beam Number | Detection mAP | Instance Seg mAP | Captioning | |
|---|---|---|---|---|
| | | | BLEU-4 | CIDEr |
| 1 | 45.6 | 40.9 | 33.0 | 108.5 |
| 2 | 47.4 | 42.1 | 34.2 | 111.1 |
| 3 | 47.8 | 42.6 | 34.0 | 110.8 |
| 5 | 47.9 | 42.6 | 33.9 | 111.0 |

Table 15: Ablation on COCO detection and instance segmentation.

| Open Ended | Beam Search | Embedding Retrieval | Copy Paste | Repeat GT | Detection AP | Ins Seg AP |
|---|---|---|---|---|---|---|
| | | | | | 45.1 | 31.4 |
| ✓ | | | | | 43.0 | 30.4 |
| ✓ | ✓ | | | | 45.1 | 31.3 |
| ✓ | ✓ | ✓ | | | 45.1 | 39.2 |
| ✓ | ✓ | ✓ | ✓ | | 46.2 | 40.4 |
| ✓ | ✓ | ✓ | ✓ | ✓ | 47.8 | 42.6 |

Table 16: Loss weight ablation.

| CE:Focal | 1:1 | 1:3 | 1:5 | 3:1 |
|---|---|---|---|---|
| Instance Seg | 43.5 | 43.7 | 43.8 | 43.4 |
| Captioning | 35.3 | 34.9 | 34.8 | 35.3 |

Table 17: Ablation of direct box prediction.

| Method | P@0.5 | FPS |
|---|---|---|
| box | 91.8 | 1.10 |
| mask2box | 90.5 | 0.57 |

# E    Inference Speed

Table 9 shows the speed of UFO-ViT-B. By using parallel decoding for multi-prediction tasks, we achieve inference speeds comparable to single-prediction tasks, despite higher resolutions ($1120^2$ vs. $224^2$) and more predictions. Table 11 shows the speed on MLLMs. Our LLaVA-1.5 variant is slower than InternVL2.5 on REC because its tokenizer converts textual numbers into longer token sequences. In embedding retrieval, the extra scaled-dot product operation only costs a negligible 0.17 ms for InternVL2.5-8B on an A100.

# F    More ablation studies

**Beam Search.** In Table 14, we present ablation studies on the beam number. As the beam number increases, performance initially improves and then stabilizes, but further increases cause a slight performance drop in the captioning. Since larger beam numbers increase inference time, we select the optimal beam number: 3 for object detection and instance segmentation and 2 for captioning.

**Loss weights.** In Table 16, we ablate loss weights on UFO-ViT-B. We jointly train both instance segmentation and image captioning tasks. Although increasing the focal loss weight is slightly better for segmentation, it leads to a drop in captioning. Since our goal is better overall multi-task performance, we set all weights to 1 to avoid adding task bias.

**Advanced training strategies.** The sparsity of positive samples in multi-prediction tasks hampers effective learning. To mitigate this, we use two advanced strategies to increase the ratio of positive samples. First, copy-paste data augmentation [23], where objects from other images are pasted onto the target image. Second, we repeat ground truth $k$ times [29], defaulting to 3. As seen in Table 15, copy-paste boosts mAP by 1.1 for detection and 1.2 for instance segmentation, while repeating ground truth further boosts mAP by 1.6 and 2.2. This demonstrates that the sparsity of is a key bottleneck, and our performance can be effectively improved with these strategies. By default, we only use these strategies for COCO detection and instance segmentation.

**Comparisons with baseline MLLMs.** In Table 18, we provide comparisons between UFO and the baseline MLLM. Firstly, UFO significantly expands the task range, enabling the model to handle all types of segmentation. Secondly, although InternVL2.5-8B can support REC and perform detection by breaking it into multiple single-category prediction tasks, its performance is markedly inferior to ours, especially in detection. This is primarily because InternVL2.5-8B cannot predict multiple boxes for a single category nor model the relationships among multiple categories, leading to insufficient and contradictory predictions. In contrast, UFO effectively supports multi-prediction tasks through local prompts, allowing it to accommodate any prediction number.

**Mask2Box.** A simple way to output box based on segmentation is `mask2box`, which uses bounding rectangle of masks. Table 17 shows the performance of `mask2box`, which is slightly lower than directly predicting boxes. This is mainly because some masks have outlier predictions, distorting the converted boxes. Moreover, boxes are generally shorter than masks, resulting in a faster speed

Table 18: Comparisons with baseline MLLMs.

| Model | Detection | Instance Seg | Semantic Seg | REC | RES | MMVP | HallBench |
|---|---|---|---|---|---|---|---|
| InternVL2.5-8B [12] | 12.5 | - | - | 90.3 | - | 76.3 | 50.1 |
| UFO-InternVL2.5-8B | 52.3 | 45.8 | 54.6 | 93.1 | 81.0 | 76.3 | 50.7 |

Table 19: DRIVE [64] Segmentation.

| Methods | 5-shot | Full fine-tuning |
|---|---|---|
| UNet++ [91] | - | 79.6 |
| ConvMixer [50] | - | 82.2 |
| GiT-H [69] | 57.9 | - |
| UFO-ViT-H | 77.4 | 82.0 |
| UFO-InternVL2.5-8B | 78.1 | 82.4 |

Table 20: Depth estimation on NYUv2 Depth [63].

| Methods | RMSE↓ | $\delta 1$↑ | REL↓ | log10↓ |
|---|---|---|---|---|
| Painter [75] | 0.327 | 0.930 | 0.090 | - |
| Unified-IO 2 [47] | 0.423 | - | - | - |
| UFO-InternVL2.5-8B | 0.305 | 0.936 | 0.087 | 0.035 |

(see Table 11). Notably, our box and mask representations are unified through the language interface. The only difference is that for boxes, textual numbers are converted into coordinates, and for masks, embedding retrieval is used. These operations are as simple as `mask2box`, which greatly reduces task-specific details compared to methods that use task decoders.

## G  Extended Experiments

**Retinal Vessel Segmentation.** Our embedding retrieval method offers superior expressive capability compared to polygons, particularly in highly complex and detailed masks, which require a large number of vertices when using polygons. To further illustrate this, we fine-tune our model on the retinal vessel segmentation, where the vessels possess very irregular and narrow shapes, which are hard to represent as polygons. We follow the few-shot settings in GiT [69], fine-tuning both UFO-ViT-H and UFO-InternVL2.5-8B on the DRIVE [64] training set for only 100 steps. In performance, UFO-ViT-H achieves 77.4 F1 score, outperforming GiT-H with 57.9 score. UFO-InternVL2.5-8B also achieves a competitive 78.1 F1 score. After fine-tuning with the entire training set, our performance can surpass strong specialized models such as UNet++ [91] and ConvMixer [50]. As shown in Figure 5, UFO accurately segments the retinal vessels. This result validates the effectiveness of our method on extremely fine-grained structures, enabling support for more general segmentation.

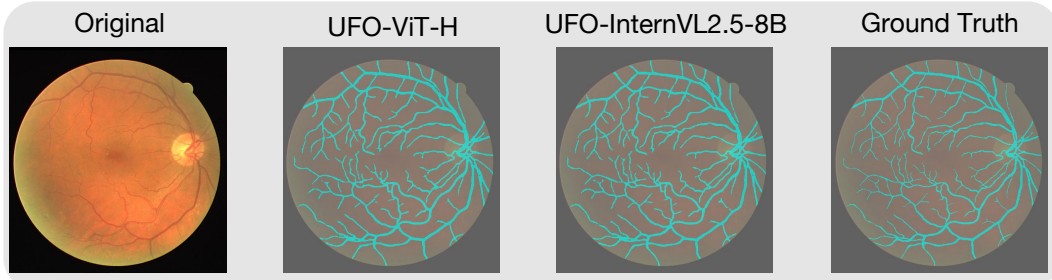

Figure 5: Visualizations of retinal vessel segmentation.

**Depth Estimation.** Thanks to the flexibility of our method, we can easily extend it to depth estimation similar to segmentation. We can apply a sigmoid to the dot product result to interpret it as relative depth $r$, which can be then mapped into absolute depth. For depth $\hat{\mathbf{D}}$ within $[\mathbf{D}_{min}, \mathbf{D}_{max}]$, we can predict it as follows:

$$r = \sigma(\frac{\mathbf{e_d}\,\mathbf{h_v}^\top}{\sqrt{d}}), \quad \hat{\mathbf{D}} = r \cdot \mathbf{D}_{max} + (1 - r) \cdot \mathbf{D}_{min} \tag{8}$$

$\mathbf{e_d}$ is the embedding of <DEPTH> token. As shown in Table 20, we can achieve competitive results.

The above approach essentially shares the same modeling as segmentation, differing only on how to interpret the model output. In segmentation, dot product results serve as confidence scores that are thresholded to create masks, whereas in depth estimation, they represent relative depth and are then converted to absolute depth. This process can be seen as a simple post-processing, which is very common in general-purpose models [75, 9]. For example, Painter [75] converts RGB values to

Table 21: Surface normal prediction on NYUv2 Depth [63]

| Method | Mean | Median | 11.25° | 22.5° | 30° |
|---|---|---|---|---|---|
| GeoNet++ [55] | 18.5 | 11.2 | 9.502 | 0.732 | 0.907 |
| Marigold [32] | 18.8 | - | 0.559 | - | - |
| GeoWizard [22] | 17.0 | - | 0.565 | - | - |
| UFO-InternVL2.5-8B | 17.8 | 10.4 | 0.543 | 0.733 | 0.800 |

categories and depth using task-specific rules. Our unification lies in modeling all tasks through the standard language interface. When specific outputs (e.g., masks, depth) are needed, the corresponding post-processing is then performed. This design unifies the core understanding capabilities across tasks while requiring only minimal, learning-free post-processing for various formats.

**Surface Normal Prediction.** Similar to depth estimation, we can extend UFO to surface normal prediction. We introduce three task-specific tokens for normal vectors: <NORMAL_X>, <NORMAL_Y>, and <NORMAL_Z>. Given the image feature $\mathbf{h}_v$ and the three directional embeddings $\mathbf{e}_x$, $\mathbf{e}_y$, and $\mathbf{e}_z$, the normal components (e.g., $\hat{n}_x$) are computed as follows:

$$\hat{n}_x = \sigma\left(\frac{\mathbf{e}_x^\top \mathbf{h}_v}{\sqrt{d}}\right)$$

where $\sigma(\cdot)$ denotes the sigmoid function and $d$ is the feature vector dimension. Finally, the predicted values are normalized to obtain a unit surface normal vector:

$$\hat{\mathbf{n}} = \frac{(\hat{n}_x,\ \hat{n}_y,\ \hat{n}_z)}{\|(\hat{n}_x,\ \hat{n}_y,\ \hat{n}_z)\|}$$

We conduct the evaluation on NYU v2 Normal Benchmark, and the performance is shown in the Table 21. It can be seen that our method achieves comparable performance to specialized models.

## H Discussions

**Comparisons with GiT.** GiT [69] also aims to build a generalist model for fine-grained perception tasks. Compared with GiT, we provide six key improvements: 1) Segmentation by embedding retrieval, a simple yet intuitive way to support segmentation by language interface. GiT uses polygons and textual classes, leading to information loss or lengthy sequences. In contrast, UFO can accurately segment using only 16 mask tokens. 2) Alignment with the open-ended language interface: unlike GiT, which requires separate vocabularies and fixed output lengths per task, UFO uses shared vocabulary and outputs arbitrary-length sequences. Tables 7 and 18 demonstrate that open-ended detection is challenging due to severe class imbalance. We address this issue with a text-aligned beam search and achieve enhanced performance. 3) Scalability to MLLMs: while GiT only experiments on relatively small ViTs, UFO can easily scale to larger MLLMs thanks to the aligned language interface. 4) Exploring the image representation capabilities of the language interface. GiT is a purely text-based method, while UFO can effectively extract mask information from image features. 5) Better task universality: GiT uses different methods for instance and semantic segmentation (polygon and textual class), while we adopt a unified approach for both tasks because UFO can support masks with any shape. As UFO is aligned with vision-language tasks, we can effortlessly combine VQA reasoning and segmentation, enabling ReasonSeg. 6) Significant performance improvements. In Table 2, UFO-ViT-H outperforms GiT-H by 1.2 mAP and 12.3 mAP on COCO detection and instance segmentation, and 3.3 mIoU on ADE20K semantic segmentation.

## I Visualization

**Multi-task Training Results.** In Figure 6, we visualize the multi-task training results of UFO-ViT-H. The model can not only handle simple perception tasks but also accurately detect and segment multiple objects in complex scenarios.

**Instruction Tuning Results.** We present qualitative results of UFO-InternVL2.5-8B in Figure 7. Leveraging the language capabilities of MLLMs, the model can accurately locate and segment based on both simple phrases and complex queries.

**Multiple Mask Tokens.** In Figure 8, we visualize the masks corresponding to each of multiple mask tokens. Each token captures specific details, such as different legs of a horse or the tail of a dog. Therefore, combining all the mask tokens results in a higher-resolution, more detailed mask. In Figure 9, we visualize the results for different numbers of mask tokens. Using only one mask token results in rough edges, while increasing the number of mask tokens produces more refined masks, leading to better performance.

**Failure cases.** We visualize the failure cases of UFO-InternVL2.5-8B on REC, RES and ReasonSeg in Figure 10. In the first image, the arm that needs to be localized is very blurry and only appears in a small portion at the edge of the image, indicating that the model has deficiencies in localizing objects with low visibility. In the second image, multiple zebras overlap and their patterns are very similar, resulting in incorrect segmentation locations, demonstrating the model's shortcomings in segmenting clustered objects. In the third example, the model fails to identify the answer as "river" due to limited knowledge, leading to segmentation errors.

# J Broader Impacts

Our approach achieves a compact model design by removing task-specific decoders and integrating diverse fine-grained perception tasks into a unified architecture. This structural efficiency reduces computational demands during training, leading to a decrease in carbon footprint. We do not identify negative social impacts currently.

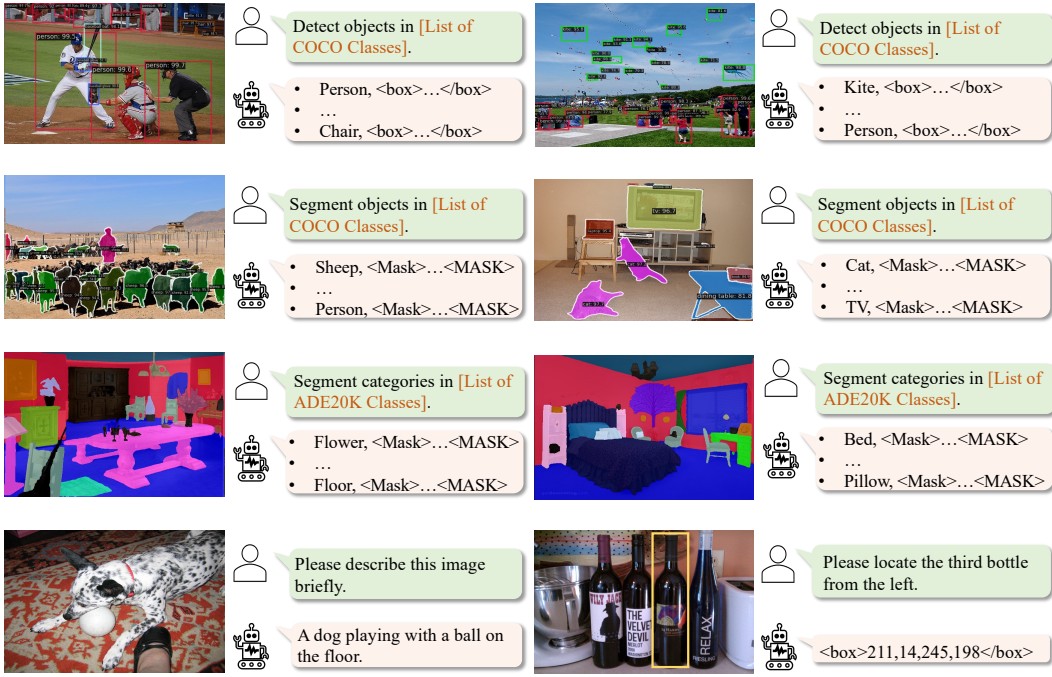

Figure 6: Qualitative results of multi-task training. The first three rows correspond to object detection, instance segmentation, and semantic segmentation, while the last row shows results on captioning and referring expression comprehension.

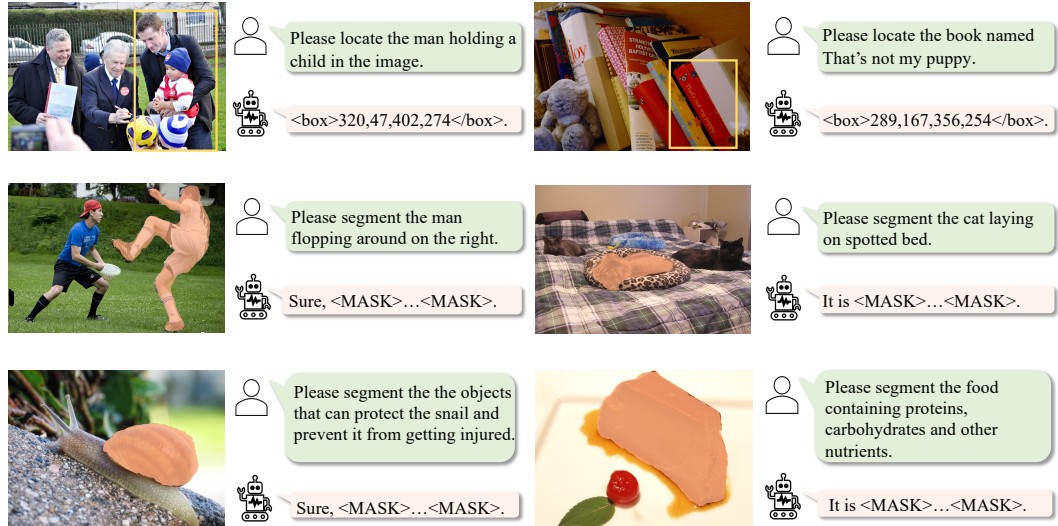

Figure 7: Qualitative results of Fine-grained Instruction Tuning. The three rows correspond to REC, RES, and reasoning segmentation in order.

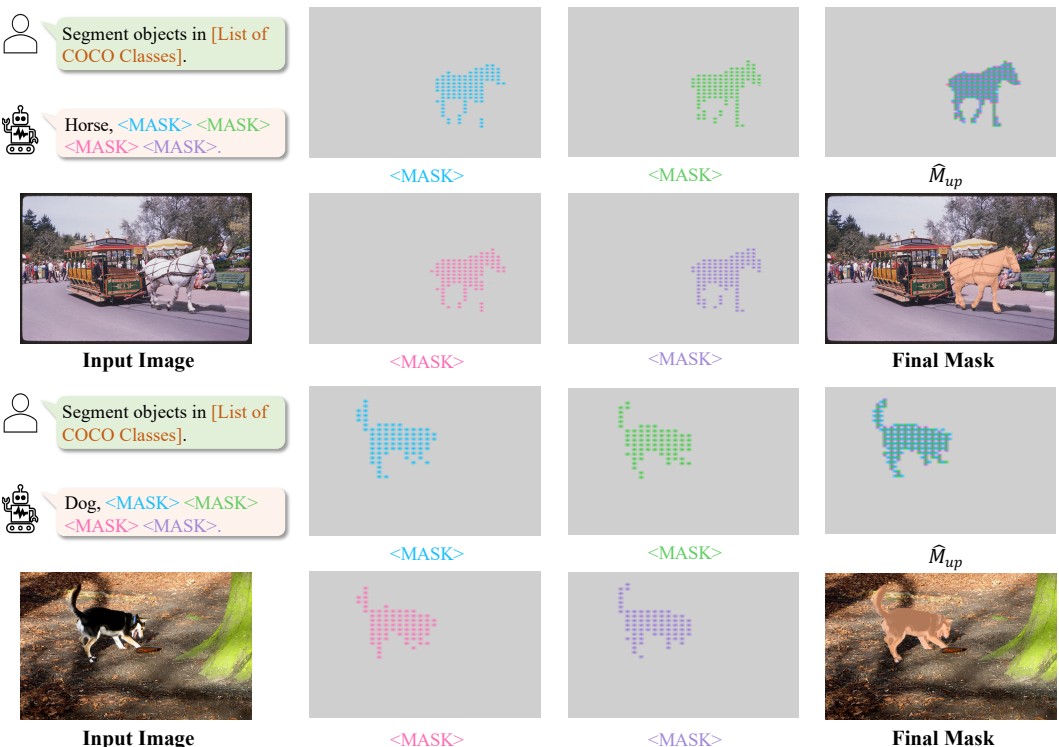

Figure 8: Visualization of multiple mask tokens. We illustrate with four mask tokens (with $N$=2). Employing multiple mask tokens allows for capturing finer details, such as the horse leg and the dog tail, resulting in more precise and refined masks.

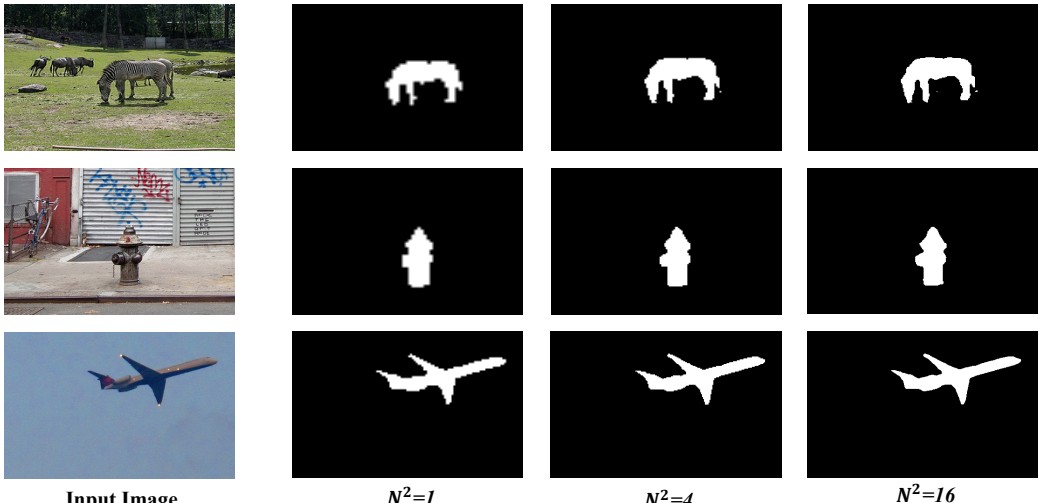

Figure 9: Segmentation results with different mask token number ($N^2$) on UFO-ViT-B$_{\text{single-task}}$.

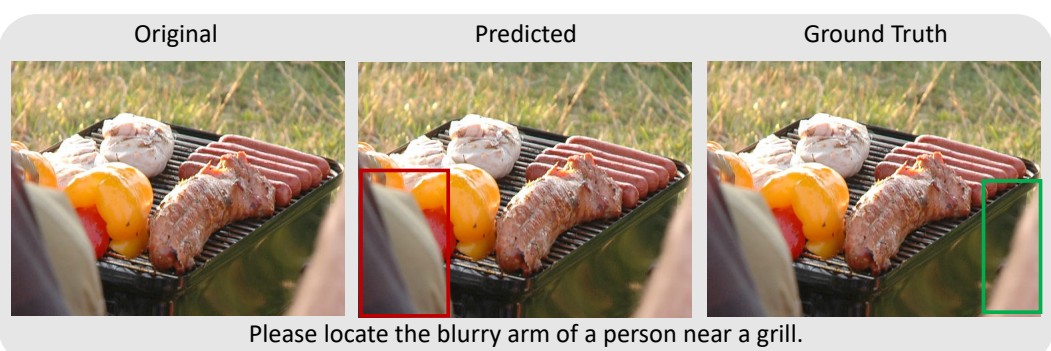

Please locate the blurry arm of a person near a grill.

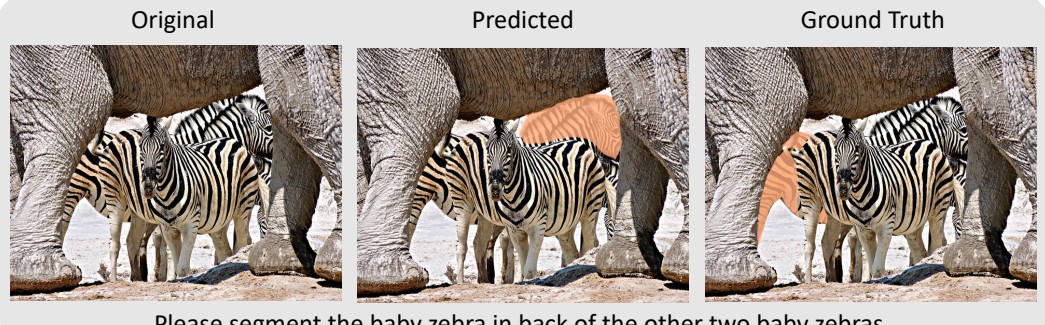

Please segment the baby zebra in back of the other two baby zebras.

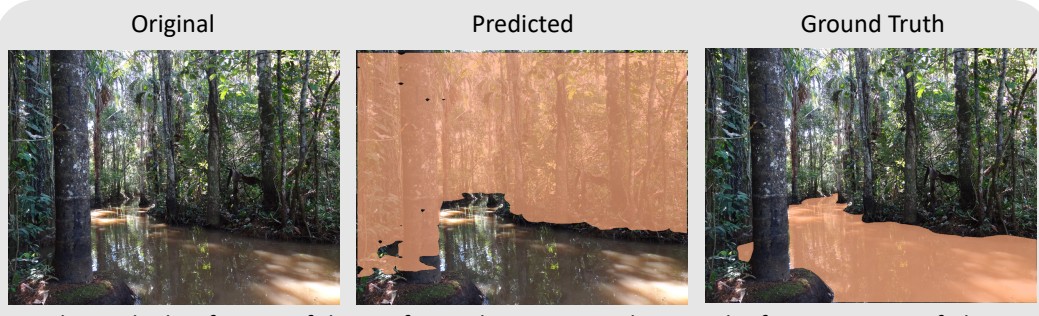

What is the key feature of the rainforest that supports the growth of various types of plants, creating a rich habitat for many animals? Please output segmentation mask.

Figure 10: Failure case visualizations of UFO-InternVL2.5-8B on REC, RES and ReasonSeg.

