# OpenReview forum: "UFO: A Unified Approach to Fine-grained Visual Perception via Open-ended Language Interface"
_NeurIPS.cc/2025/Conference — NeurIPS 2025 spotlight_

### Official Review · Reviewer_vFSU · 2025-06-26

**Clarity:** 3
**Significance:** 3
**Originality:** 3
**Rating:** 5
**Confidence:** 4

**Summary:**

This paper introduces UFO, a novel framework that unifies various fine-grained visual perception tasks. The core motivation is to better align the representation of such tasks with an open-ended language interface, enabling seamless integration with multimodal large language models (MLLMs) without the need for auxiliary modules. To support segmentation, UFO reformulates the task as an embedding retrieval problem, generating segmentation masks based on the similarity between mask embeddings and visual features. Additionally, the paper proposes a multi-token prediction strategy that effectively refines the masks. Experimental results demonstrate that UFO achieves strong performance across multiple benchmarks, with significant improvements observed on COCO, ADE20K, and ReasonSeg.

**Questions:**

1.	In the segmentation task, the visual features are obtained from the outputs of the LLMs. Have the authors explored using the visual features directly from the Vision Transformer within the MLLM instead? I am interested in understanding how the visual features change before and after being processed by the LLM.
2.	In open-ended decoding tasks like object detection, could the generation of synonyms for COCO categories potentially affect the evaluation metrics? Additionally, what is the method used to compute the detection score for each predicted bounding box?
3.	The paper employs a method that directly predicts discrete text tokens for object detection. I am curious whether an alternative approach—inferring bounding boxes by computing the minimum enclosing rectangles from the predicted masks—would be a feasible and effective solution.

**Ethical Concerns:**

["NO or VERY MINOR ethics concerns only"]

**Final Justification:**

Thanks for the rebuttal, I am sticking to my original positive rating.

**Limitations:**

YES

**Quality:**

3

**Strengths And Weaknesses:**

Strengths
1.	A unified framework for visual tasks is essential for the development of multimodal foundation models. This work successfully integrates fine-grained visual perception with vision-language understanding, substantially reducing both architectural and training complexity while leveraging cross-task synergies.
2.	The embedding retrieval approach to segmentation is innovative. It effectively leverages the previously underexploited visual representation capabilities of MLLMs. Additionally, the proposed mask up-sampling method is compelling, allowing for precise mask refinement using only a minimal number of tokens.
3.	The method is thoroughly evaluated across various tasks, demonstrating competitive performance and clear superiority.
Weaknesses:
Some technical details and ablations are not sufficiently clear. See questions below.

---

> ### Author Rebuttal · Authors · 2025-07-27
>
> Dear Reviewer vFSU,
>
> Many thanks for your recognition of our work. We will address your questions one by one below.
>
> > [Q1] Have the authors explored using the visual features directly from the Vision Transformer within the MLLM instead?
>
> Thanks for asking about this important design. We have conducted comprehensive experiments to compare the differences between using ViT features and using visual features from the LLM. As shown in the table below, relying on ViT features results in a significant drop in performance. We attribute this to the following reasons:
>
> 1. Compared to ViT features, the visual features output by the LLM are better aligned with the feature space of the mask embeddings.
> 2. The LLM-generated visual features are more discriminative. Because we **place the text before the image**,  the LLM can attend to the text prompt when processing the visual features. As a result, LLM can **dynamically emphasize image regions relevant to the prompt**, which helps the mask embeddings better localize the target. Reversing the input order (image before text, second row in the table) will also lead to a performance drop, highlighting the importance of contextualizing visual features with textual information.
>
> | ViT | LLM | Text First |  RES  | ReasonSeg |
> |:---:|:---:|:----------:|:-----:|:---------:|
> |  ✓  |     |            | 74.2  |   55.4    |
> |     |  ✓  |            | 77.9  |   56.7    |
> |     |  ✓  |     ✓      | 80.0  |   60.0    |
>
>
>
> > [Q2]  Could the generation of synonyms for COCO categories potentially affect the evaluation metrics? What is the method used to compute the detection score?
>
> In the evaluation, we find that all categories predicted by the model are within the 80 classes provided by COCO, and there are no cases of predicting synonyms of the categories. This is mainly because we specify the category names in the text prompt, and the model is able to follow the prompt well.
>
> We use the softmax logit of the first token predicted in each text sequence as the confidence score for the entire prediction. For example, for the sequence "Duck, `<box>`...", we use the score associated with the "Duck" token.
>
> > [Q3] Whether inferring bounding boxes by computing the minimum enclosing rectangles from the predicted masks is a feasible and effective solution.
>
> Yes, predicting bounding boxes using the `mask2box` approach is feasible, and we have also included this ablation in Appendix F. The performance of UFO-ViT-B on the COCO Detection using this method is presented in the table below. The performance is slightly lower than predicting the textual box. This is mainly because some low-confidence masks have outlier predictions, leading to distorted boxes. The inference speed is also slower, as the mask sequences are generally longer than the box. Therefore, to achieve better accuracy and efficiency, we retain the method of predicting text coordinates for detection tasks.
>
> | Method   | AP | FPS  |
> | -------- | ----- | ---- |
> | Textual Box      | 47.8  | 4.1 |
> | Mask2Box | 45.6 | 3.6 |
>
> ***
>
> Thanks for your professional comments, we will add these experiments and details to our revision. Feel free to let us know if you have any further questions or concerns :-).

---

> > ### Comment · Reviewer_vFSU · 2025-08-06
> >
> > Thanks for the rebuttal, I am sticking to my original positive rating.

---

> > > ### Author Response · Authors · 2025-08-06
> > > **Thanks for your time, effort, and kind words!**
> > >
> > > Thanks for your acknowledgement on our paper and rebuttal. We will remain active until the discussion ends. Please feel free to get back to us if you have any new questions :-)!

---

### Official Review · Reviewer_99r1 · 2025-06-27

**Clarity:** 3
**Significance:** 3
**Originality:** 3
**Rating:** 5
**Confidence:** 4

**Summary:**

This paper introduces UFO, a framework designed to unify fine-grained visual perception tasks (object detection, segmentation) and standard vision-language tasks within a single model, using an open-ended language interface. The key problem it addresses is the architectural and training complexity that arises from adding task-specific decoders to Multimodal Large Language Models (MLLMs). UFO's core contributions are twofold: 1) It formulates all perception outputs as text sequences, representing boxes as textual coordinates and, more notably, 2) It introduces a novel "embedding retrieval" mechanism for segmentation.

**Questions:**

1.Considering that UFO's training includes interactions with image tokens, how does this additional supervision impact the model's perception and understanding capabilities? Could further ablation studies be conducted to explore this? For instance, what would happen if a detach operation were applied to the image tokens?

2. Can the embedding retrieval approach also be applied to detection tasks? Specifically, could box embeddings be obtained through interactions with image tokens, similar to traditional anchor-based methods?

**Ethical Concerns:**

["NO or VERY MINOR ethics concerns only"]

**Final Justification:**

Thank you to the authors for the detailed and thoughtful rebuttal. I appreciate the clear clarifications provided regarding the open-ended language interface, the inference and evaluation process, and the post-processing pipeline. I encourage the authors to incorporate the clarifications provided in the rebuttal into the final version of the paper for greater clarity.
 Based on the improved understanding from the rebuttal, I am raising my score to a 5.

**Limitations:**

See Weakness.

**Quality:**

3

**Strengths And Weaknesses:**

Strengths:
1. The paper tackles a timely and significant problem: the integration of fine-grained perception into general-purpose MLLMs. The proposed decoder-free approach is a notable departure from prevailing methods that rely on adding complex, task-specific heads , which simplifies the overall architecture and training pipeline.
2. The "segmentation by embedding retrieval" is a particularly clever and intuitive idea. Leveraging the model's own output embeddings to query rich image features is a powerful concept that sidesteps the information loss issues of polygon approximations or the inefficiency of per-pixel text classification.
3. The experiments are comprehensive, covering a wide range of tasks and datasets, which strengthens the paper's contributions and demonstrates the framework's versatility.
4. The paper is well-written and structured, making it accessible to a broad audience, including those who may not be specialists in the field.


Weaknesses:
1. About 'open-ended language interface': The paper does not provide a clear definition or explanation of what constitutes an "open-ended language interface."
After all, the current method still requires learning to output implicit tokens and involves additional interactions with image tokens. In my view, methods like SAM4MLLM[1] or SegAgent[6], which only require outputting pure text, might better align with the definition of an "open-ended language interface."


2. The paper does not provide sufficient details on the inference and evaluation processes for object detection and instance segmentation tasks. Specifically, how does the method handle multiple objects and categories? How many forward passes are required? Are there any additional post-processing steps? These details are crucial for understanding the practical applicability and scalability of the method.

3. The comparison methods are relatively old and may not fully represent the current state-of-the-art in the field. Taking the RES task as an example, some newer and stronger works have not been included in the comparison, such as:
Ground-V[2], OMG-LLAVA[3], POPEN[4], F-LMM[5], SegAgent[6], HyperSeg[7].

[1]SAM4MLLM: Enhance Multi-Modal Large Language Model for Referring Expression Segmentation.      ECCV2024

[2] Ground-V: Teaching VLMs to Ground Complex Instructions in Pixels.  CVPR2025

[3] Omg-llava: Bridging image-level, object-level, pixel-level reasoning and understanding.       Neurips2024

[4]Popen: Preference-based optimization and ensemble for lvlm-based reasoning segmentation.        CVPR2025

[5]F-LMM: Grounding Frozen Large Multimodal Models.    CVPR2025

[6]Segagent: Exploring pixel understanding capabilities in mllms by imitating human annotator trajectories.     CVPR2025

[7]HyperSeg: Hybrid Segmentation Assistant with Fine-grained Visual Perceiver.    CVPR2025

---

> ### Author Rebuttal · Authors · 2025-07-27
>
> Dear Reviewer 99r1,
>
> We sincerely appreciate your acknowledgment of the **significance, intuition, thorough experiments, and clarity** of our paper. We will respond to your questions one by one.
>
> > [W1] Explanation of open-ended language interface.
>
> Thanks for pointing out this important concept. In the paper, we refer to the "open-ended language interface" as the capability to **output variable-length text sequences, with termination indicated by the `<END>` token.** This stands in contrast to prior works such as GiT [1], which are limited to fixed-length outputs.
>
> Our method does not conflict with open-ended language interfaces.
>
> 1. Although we require fixed-length `<MASK>` tokens for segmentation, we still perform text generation in an open-ended manner. **Only after text generation is complete**, we check whether the output contains `<MASK>` segments of the required length.
> 2. The interaction with image features is also carried out only after the text generation is finished.
>
> Notably, after the open-ended text generation ends, **our segmentation mask is determined by the image features and mask embeddings**. In contrast,  SAM4MLLM and SegAgent can only produce coarse prompts by text generation, which requires further refinement by SAM. We will clarify this term in our revision.
>
>
> [1] GiT: Towards Generalist Vision Transformer through Universal Language Interface. ECCV 2024
>
> > [W2] Details on the inference and evaluation processes for object detection and instance segmentation tasks.
>
> > [W2.1]  How does the method handle multiple objects and categories?
>
> We handle this by the **parallel decoding described in Section 3.4**, which is also illustrated in Figure 3.  During inference,  we sample grid points over the image of size $K\times K$, resulting in a total of $M = K^2$ points. **Each grid point serves as a local prompt and is responsible for detecting its nearby object**. We interpolate the image features at each grid location to extract the grid features: `grid_feats = torch.nn.functional.grid_sample(image_feats, grid_locations)`
>
> We then input the text prompt, the image features, and the grid features into the LLM. Take the rightmost image in Figure 3 as an example:
> $$
> \text{Detect cow, person, duck.}<\mathrm{Image}><\mathrm{Grid}_1><\mathrm{Grid}_2>\ldots<\mathrm{Grid}_M>
> $$
>
> `<Grid>` refers to the `<Local>` token in Section 3.4. To ensure independence among $M$ predictions, we adjust the attention mask so that each grid feature is isolated from the others. Then we start generating in an autoregressive manner, with the difference that **we predict $M$ tokens at each forward** instead of just one.
>
> $$
> <\mathrm{Grid}_1><\mathrm{Grid}_2>\ldots<\mathrm{Grid}_M> | <T^1_1><T^1_2>\ldots<T^1_M> | <T^2_1><T^2_2>\ldots<T^2_M> | \ldots
> $$
>
> where $T^j_i$ denotes the $j$-th token in the $i$-th sequence and `|`  distinguishes the tokens of each forward pass. When all $M$ sequences have predicted the `<END>` token, decoding ends. Finally, we obtain M sequences with different objects and categories, which might look like: (1) Duck, `<box>`... (2) Background $\cdot \cdot \cdot$ (M) Cow, `<box>`...
>
> For instance segmentation, the process remains the same except that the textual boxes are replaced with `<MASK>`  tokens.
>
>
>
> > [W2.2]   How many forward passes are required?
>
> As explained above, only $1+L$ forward passes are required, where $L$ is the maximum length of the output sequence. Typically, for detection, $L$ is less than 20, and for segmentation, $L$ is less than 30. Thanks to our parallel decoding strategy, the number of forward passes is **independent of the number of objects in the image**, which greatly accelerates inference. For example, in the semantic segmentation, UFO-ViT-B achieves 4.8 FPS, which is 3$\times$ faster than SAM-B (1.6 FPS). More speed results are in Appendix Table 9.
>
>
>
> > [W2.3]  Are there any additional post-processing steps?
>
> Yes, the post-processing mainly consists of the following steps:
>
> 1. Pattern matching:  We need to extract class names and box coordinates from text sequences.
> 2. Extract the confidence scores. We use the softmax logit of the first token predicted as the confidence score for the prediction. For example, for the sequence "Duck, `<box>`...", we use the score associated with the "Duck" token.
> 3. Finally, we apply NMS to filter boxes and masks.
>
>
>
> > [W3] Some newer and stronger works are not included.
>
> Many thanks for listing these outstanding works. We compare our method with these approaches on the RES and ReasonSeg below. UFO achieves a highly competitive performance on RES, outperforming all methods except HyperSeg on average. On ReasonSeg, UFO achieves the **best** performance, surpassing HyperSeg by **4.3** gIoU on the validation set and Ground-V by **14.9** gIoU on the test set. These results show that UFO unifies reasoning and segmentation more effectively. We will add these comparisons in the revision.
>
> |  Methods  |      | RefCOCO |       |      | RefCOCO+ |       | RefCOCOg |      | Average | ReasonSeg |      |
> | :-------: | :--: | :-----: | :---: | :--: | :------: | :---: | :------: | :--: | :-----: | :-------: | :--: |
> |           | val  |  testA  | testB | val  |  testA   | testB |   val    | test |         |    val    | test |
> | Ground-V  | 83.9 |  85.0   | 82.1  | 73.1 |   75.8   | 69.8  |   74.8   | 74.2 |  77.3   |     -     | 52.1 |
> | OMG-LLaVA | 77.2 |  79.8   | 74.1  | 68.7 |   73.0   | 61.6  |   71.7   | 71.9 |  72.3   |     -     |  -   |
> |   POPEN   | 79.3 |  82.0   | 74.1  | 73.1 |   77.0   | 65.1  |   75.4   | 75.6 |  75.2   |   60.2    |  -   |
> |   F-LMM   | 76.1 |    -    |   -   | 65.2 |    -     |   -   |   68.5   |  -   |    -    |   46.7    | 46.2 |
> | SegAgent  | 79.7 |  81.4   | 76.6  | 72.5 |   75.8   | 66.9  |   75.1   | 75.2 |  75.4   |     -     |  -   |
> | HyperSeg  | 84.8 |  85.7   | 83.4  | 79.0 |   83.5   | 75.2  |   79.4   | 78.9 |  81.2   |   59.2    |  -   |
> |    UFO    | 81.0 |  82.6   | 78.6  | 77.1 |   80.4   | 72.6  |   76.7   | 77.3 |  78.3   |   63.5    | 67.0 |
>
>
> &nbsp;
> > [Q1] Ablation of supervision of image tokens.
>
> Thanks for this insightful question. We explore the impact of applying supervision to image tokens on UFO-InternVL2.5-8B:
>
>
> | Supervision | REC      | RES      | ReasonSeg | MMBench  | HallusionBench |
> |:---:|:---:|:----------:|:-----:|:---------:|:---------:|
> |           | 91.6     | 79.2     | 58.8      | **84.4** | 50.3           |
> | ✓         | **91.8** | **80.0** | **60.0**  | 84.2     | **50.7**       |
>
>
>
> As shown in the table, supervising image tokens can largely improve segmentation while maintaining original capabilities. We attribute this to two reasons:
>
> 1. InternVL2.5-8B does not explicitly supervise image tokens during training; instead, it relies on sparse textual signals, which leads to insufficient preservation of fine details (e.g., object boundaries) in the visual features. Therefore, directly using these visual features can lead to degraded segmentation performance.
> 2. The supervision in the segmentation provides fine-grained and direct signals that help preserve richer details in the visual features, thereby enhancing their discriminability.
>
>
>
> > [Q2] Can the embedding retrieval approach also be applied to detection tasks? For example, obtain box embeddings through interactions with image tokens.
>
> Yes, there are two different implementations:
>
> 1. Mask2Box: Convert the mask into a box by extracting its minimum enclosing rectangle. Its performance is slightly lower than predicting textual box. This is mainly because some low-confidence masks have outlier predictions,  leading to distorted boxes. The inference speed is also slower, as the mask sequences are generally longer than box.
> 2. Special Token: Predicting box coordinates by interactions with image features.  Specifically, we still decompose the detection task into multiple sub-tasks, each corresponding to a grid point. Each point predicts the textual class and four special tokens (`<X1>`, `<Y1>`, `<X2>`, `<Y2>`) representing the box offsets relative to the point. Each offset (e.g., $\Delta x_1$) is computed using the scaled dot product between the token embedding  $ e_{X1}$ and the grid feature $h_g$, followed by a sigmoid:
>
> $$
> \begin{align*}    \Delta x_1 &= \sigma\left(\frac{e_{X1}^\top h_g}{\sqrt{d}}\right)   \end{align*}
> $$
>
>
> where $ \sigma(\cdot)$ is the sigmoid function and $d$  is the dimension.  As grid features are interpolated from image features, this can be seen as extracting box information from the image.
>
> This approach is faster due to the shorter sequence, but its performance is worse than textual boxes. There may be two reasons: 1. Shorter sequences are less expressive. 2. This approach must store all offset information in a single grid feature, while the original method can predict textual boxes by attending to both image and grid features.
>
> | Method        | AP   | FPS  |
> | :------------ | ---- | ---- |
> | Textual Box   | 47.8 | 4.1  |
> | Mask2Box      | 45.6 | 3.6  |
> | Special Token | 43.7 | 4.5  |
>
>
>
> ***
>
> Thanks again for your professional, detailed, and valuable reviews! We have done our best to address each of your concerns and hope our response can resolve them. Please let us know if you have any other questions. We will actively join the discussion until the end of the rebuttal period. We are looking forward to hearing from you :-)!

---

> > ### Comment · Reviewer_99r1 · 2025-08-04
> >
> > Thank you to the authors for the detailed and thoughtful rebuttal. I appreciate the clear clarifications provided regarding the open-ended language interface, the inference and evaluation process, and the post-processing pipeline. I encourage the authors to incorporate the clarifications provided in the rebuttal into the final version of the paper for greater clarity.
> >  Based on the improved understanding from the rebuttal, I am raising my score to a 5.

---

> > > ### Author Response · Authors · 2025-08-04
> > > **Thanks for your timely reply and the kind words!**
> > >
> > > We're glad to hear that your concerns have been addressed. We will carefully incorporate the clarifications from our rebuttal into the final version of the paper to further improve its clarity. Thank you again for your constructive suggestions!

---

### Official Review · Reviewer_ZRJs · 2025-07-02

**Clarity:** 3
**Significance:** 3
**Originality:** 3
**Rating:** 5
**Confidence:** 4

**Summary:**

The paper presents UFO, a novel framework that integrates fine-grained visual perception tasks such as object detection, segmentation, and image-level vision-language tasks into a single model using an open-ended language interface. The key innovation is the embedding retrieval approach for segmentation tasks, which leverages the language interface to generate masks without task-specific decoders. The authors demonstrate significant improvements over previous state-of-the-art models, particularly in instance and semantic segmentation tasks.

**Questions:**

See Strengths And Weaknesses

**Ethical Concerns:**

["NO or VERY MINOR ethics concerns only"]

**Final Justification:**

The authors have addressed my concerns, I keep my rating.

**Limitations:**

Yes

**Quality:**

3

**Strengths And Weaknesses:**

Strengths

1. The paper introduces a unified approach to handle diverse fine-grained visual perception tasks through a single model, which is a significant step towards more general and versatile multimodal models. This approach simplifies architectural design and training strategies.

2. The embedding retrieval method for segmentation is effective. It leverages the inherent capabilities of MLLMs to generate precise masks without relying on complex task-specific architectures, leading to improved performance and reduced computational complexity.

3. The proposed method outperforms previous state-of-the-art models on multiple benchmarks, including COCO instance segmentation and ADE20K semantic segmentation. The results demonstrate the effectiveness of the unified approach in achieving high accuracy and robustness.

4. The framework is shown to be scalable and easily integrable with existing MLLMs, such as InternVL2.5 and LLaVA-1.5. This scalability allows the model to benefit from advanced language capabilities, enhancing its performance on tasks that require reasoning and fine-grained perception.

Weaknesses

One potential concern is the impact of integrating fine-grained perception tasks on the model's general question-answering (Q&A) performance. While the paper demonstrates strong performance on specific tasks, it is unclear how these specialized tasks might conflict with or affect the model's ability to handle broader, more general Q&A tasks.

---

> ### Author Rebuttal · Authors · 2025-07-27
>
> Dear Reviewer ZRJs,
>
> Many thanks for recognizing the **novelty, versatility, effectiveness, strong performance and scalability** of UFO. We address your questions as follows.
>
> > [W1] How these specialized tasks might conflict with or affect the model's ability to handle broader, more general Q&A tasks?
>
> Thanks for pointing out this important concern. We have included the performance on general QA in Table 5 of the main paper, and we also paste the table here for convenience.
>
> | Models             | GQA  | MMBench | MMVP | HallusionBench |
> | ------------------ | ---- | ------- | ---- | -------------- |
> | InternVL2.5-8B     | 60.6 | 84.6    | 76.3 | 50.1           |
> | UFO-InternVL2.5-8B | 60.8 | 84.2    | 76.3 | 50.7           |
>
> As shown in the table, UFO achieves performance on par with the baseline MLLM on VQA benchmarks. We attribute this to three key factors:
> 1. We employed a relatively small LoRA rank (set to 8), which minimizes modifications with the pretrained MLLM parameters.
> 2. Our training corpus contains a substantial proportion (40%) of high-quality VQA data from LLaVA-OneVision [1], helping to preserve and reinforce general QA performance. Additionally, we observed a 0.6-point improvement on HallusionBench, which may indicate that our fine-grained fine-tuning helps reduce hallucinations.
> 3. UFO is highly aligned with general QA tasks, both adopting an open-ended language interface. As discussed in Lines 147-149 of the main paper, we argue that since the model can already answer where and what objects are in general QA, the mask information is already encoded in the image features. Our method can decode this information and further enhance it through training, thus essentially sharing the same capabilities as general QA tasks.
>
> [1] LLava-Onevision: Easy visual task transfer. Arxiv 2024.
>
> ***
> We hope the above response can help solve your questions. Thanks again for your thorough review and looking forward to your reply :-) !

---

### Official Review · Reviewer_nJnP · 2025-07-20

**Clarity:** 2
**Significance:** 3
**Originality:** 3
**Rating:** 5
**Confidence:** 2

**Summary:**

This paper proposes a unified framework for fine-grained perception tasks (like detection and segmentation) by expressing any perception task’s output as open-ended texts. The goal is to remove the reliance on task-specific modules like mask decoders. For detection, the boxes are expressed as textual numbers, while segmentation is formulated as embedding retrieval problem. To do so, a $<$MASK$>$ token is added to the vocabulary to indicate the segmentation task, and its embedding is used as query to retrieve the image features corresponding to the mask token. Also for segmentation, instead of using a single mask token and then upsample by interpolation, the authors propose predicting multiple mask tokens and using each of them to query visual features. Experiments are conducted on multi-task training and fine-grained instruction tuning.

**Questions:**

* From Fig.1, it’s not clear how detection is different between (b) and (c). Could the authors please explain what is the advantage of predicting box location as text in UFO compared to predicting location tokens as in (b)?

**Ethical Concerns:**

["NO or VERY MINOR ethics concerns only"]

**Final Justification:**

The authors fully addressed my concerns, therefore I raise my score to 5. I encourage the authors to include the normal prediction experiments, as the results look strong compared to state-of-the-art methods like Marigold. I also encourage the authors including the trainable parameters comparison.

**Limitations:**

yes

**Quality:**

3

**Strengths And Weaknesses:**

**Strengths**

* Casting perception targets as open-ended language sequences is intriguing as an idea to unify visual tasks and integrate more complicated fine-grained tasks like segmentation. This unification is an important step to get closer to the ultimate goal of a generalist model.

* In multi-task training, UFO outperforms both specialist models and previous generalist models, and can seamlessly integrate to MLLMs.

* UFO achieves SOTA results on visual grounding and reasoning segmentation.


**Weaknesses**

* Integrating fine-grained perception tasks like detection and segmentation to generalist models under a unified open-ended text generation framework is very interesting, however it doesn't include other visual tasks. For instance, is it possible to integrate under the same framework tasks like estimating scene flow, optical flow or normals?

* The paper lacks a comparison of the methods in terms of the number of trainable parameters when applying LoRA to train UFO.

* While UFO outperforms specialist models, the latter have less parameters, and in practical cases one might prefer using a specialist model for inference speed considerations.

Minor:

* Figure 1(a) illustrates VisionLLM v2 while the text mentions LISA (Lines 33-34).
* Lines 82-83: (3) is not a contribution but a consequence of the proposed method.
* Table 1 caption: “We construct three variants by this formulation.” $\rightarrow$ this sentence is unclear

Typos:
* Line 126: Tabel -> Table
* Line 245: entrich -> enrich
* Lines 247: Tabel -> Table

---

> ### Author Rebuttal · Authors · 2025-07-27
>
> Dear Reviewer nJnP,
>
> Many thanks to your valuable comments and questions, which help us a lot to improve our work. We address your questions as follows.
>
> > [W1] Is it possible to integrate other visual tasks like estimating scene flow, optical flow or normals under the same framework?
>
> Thanks for this insightful question regarding the generalizability of UFO. Our framework can naturally extend to more tasks, such as depth estimation and surface normal prediction. The core insight is to **regard embedding retrieval as a general approach for extracting information from image features**. While we focus on mask information extraction in the main paper, the same approach readily applies to other modalities (e.g., depth or surface normals) by introducing corresponding special tokens to interact with image tokens. We have discussed this extension to **depth estimation in Appendix G**, along with performance on NYUv2 Depth. For your convenience, the comparison table is also included below.
>
> | Method             | RMSE ↓    | δ1↑       | REL ↓     |
> | ------------------ | --------- | --------- | --------- |
> | Painter [1]          | 0.327     | 0.930     | 0.090     |
> | Unified-IO 2 [2]       | 0.423     | -         | -         |
> | UFO-InternVL2.5-8B | **0.305** | **0.936** | **0.087** |
>
> For the **surface normal** prediction task, we can train the model to predict three task-specific tokens:  `<NORMAL_X>`, `<NORMAL_Y>`, and `<NORMAL_Z>`.  Given the image feature $\mathbf{h}_v$ and three embeddings $\mathbf{e}_x$, $\mathbf{e}_y$, and $\mathbf{e}_z$ for each direction, the normal value (e.g., $\hat{n}_x$) is computed as follows:
>
> $$
> \begin{align*}
> n_x &= \sigma\left(\frac{\mathbf{e}_x^\top \mathbf{h}_v}{\sqrt{d}}\right), \hat{n}_x =2\cdot n_x-1
> \end{align*}
> $$
>
> where $\sigma(\cdot)$ denotes the sigmoid function and $d$ is the feature dimension. Finally, the predicted values are normalized to obtain a unit surface normal vector:
>
> $$
> \mathbf{\hat{n}} = \frac{(\hat{n}_x,\, \hat{n}_y,\, \hat{n}_z)}{\left\|(\hat{n}_x,\, \hat{n}_y,\, \hat{n}_z)\right\|}
> $$
> We conduct the evaluation on NYUv2 Normal Benchmark, and the performance is shown in the table below. UFO can achieve comparable performance to specialized models.
>
> | Method             | Mean↓ | Median↓ | 11.25°↑ | 22.5°↑ | 30°↑  |
> | ------------------ | ----- | ------- | ------- | ------ | ----- |
> | GeoNet++ [3]       | 18.5  | 11.2    | 0.502   | 0.732  | 0.807 |
> | Marigold [4]       | 18.8  | -       | 0.559   | -      | -     |
> | GeoWizard [5]      | 17.0  | -       | 0.565   |-    |-      |
> | UFO-InternVL2.5-8B | 17.8  | 10.4    | 0.543   | 0.733  | 0.800 |
>
>
> A possible approach for scene flow and optical flow is to use special tokens to capture motion information from video features. However, as video training is more computationally intensive, we leave this for future work. Extending UFO to videos is highly valuable, and we will highlight this in the future work section and actively explore it!
>
> [1] Painter: Images speak in images: A generalist painter for in-context visual learning. CVPR 2023
>
> [2] Unified-IO 2: Scaling Autoregressive Multimodal Models with Vision, Language, Audio, and Action. CVPR 2024
>
> [3] GeoNet++: Iterative Geometric Neural Network with Edge-Aware Refinement for Joint Depth and Surface Normal Estimation. TPAMI 2020
>
> [4] Repurposing Diffusion-Based Image Generators for Monocular Depth Estimation. CVPR 2024
>
> [5] GeoWizard: Unleashing the Diffusion Priors for 3D Geometry Estimation from a Single Image. ECCV 2024
>
> &nbsp;
>
> > [W2] Comparison of trainable parameters when applying LoRA to train UFO.
>
> | Method             | Base (M)LLM    | LoRA Rank | LoRA Parameters | Text Embedding |
> | ------------------ | -------------- | --------- | --------------- | -------------- |
> | Cores-7B           | LLaVA-7B       | 8         | 20M             | 262M           | 641M |
> | GLaMM-7B           | Vicuna-7B      | 8         | 20M             | 262M           | 641M |
> | SAM4MLLM-8B        | Qwen-VL-7B     | 256       | 693M            | 1.24B          | 641M |
> | UFO-LLaVA-1.5-7B   | LLaVA-1.5-7B   | 8         | 20M             | 262M           | /    |
> | UFO-InternVL2.5-8B | InternVL2.5-8B | 8         | 19M             | 758M           | /    |
>
> During LoRA training, our trainable parameters include both the LoRA parameters and text embeddings. As shown in the table, UFO’s LoRA parameter count is comparable to other models and much smaller than that of SAM4MLLM. Unlike other methods, which also require training a mask decoder or even the entire LLM (e.g., HiMTok-8B, VisionLLMv2, VistaLLM), UFO achieves **superior performance with fewer trainable parameters**. We will add this comparison to the revision to further emphasize our advantages.
>
>
>
>
>
> > [W3] Although UFO outperforms specialist models, they have fewer parameters and may be preferred for faster inference.
>
> Thanks for this insightful comment. The choice between UFO and specialist models depends on scenario requirements.
>
> Specialist models are ideal for single, well-defined tasks (e.g., closed-set detection) where complex textual understanding isn't needed, as they can offer both strong performance and fast inference. In contrast, generalist models like UFO excel in complex or dynamic scenarios that require handling **multiple tasks** or **advanced reasoning over textual instructions**. As shown in Table 4 of the main paper, specialist models perform poorly on reasoning segmentation tasks compared to MLLM-based approaches.
>
> In summary, UFO and specialist models are complementary: UFO is suited for scenarios requiring general capabilities and complex reasoning, while specialist models are preferable for straightforward, single-task problems without the need for intricate textual understanding.
>
>
>
> > [W4] Unclear writing and typos.
>
> Thanks for these detailed observations! In Table 1, “We construct three variants by this formulation” refers to the three UFO model variants: UFO-ViT, UFO-LLaVA-1.5, and UFO-InternVL2.5-8B, each consisting of an Image Tokenizer, Text Tokenizer, and Multimodal Transformer. We have revised the manuscript to clarify this and correct all ambiguities and typos.
>
>
>
> > [Q1] Advantage of predicting box location as text compared to predicting location tokens.
>
> Thanks for this interesting question. The main advantage of representing box coordinates with text is its simplicity and flexibility.
>
> 1. Using text representation can avoid adding thousands of new tokens to the text tokenizer, which is easier to implement and better aligned with the original MLLM.
> 2. If a finer coordinate granularity is needed (e.g., increasing the discrete intervals from 1,000 to 2,000), text representation allows direct reuse of prior pre-training, while location tokens would need re-initialization, making adaptation less efficient.
>
> ***
>
> Thanks again for helping us improve the paper and hope our response can resolve your concerns! Please let us know if you have any further questions. We will be actively available until the end of rebuttal period. Looking forward to hearing from you :-) !

---

### Decision · Program_Chairs · 2025-09-17

**Decision:**

Accept (spotlight)

**Comment:**

This paper initially received generally positive scores. The authors have submitted a rebuttal, and after considering it, all reviewers expressed satisfaction with the responses and updated their scores to acceptance (score 5). The reviewers also encourage the authors to incorporate the clarifications and experiments provided in the rebuttal into the final version. The AC concurs with the reviewers that this is a solid submission with advantages of strong performance, novel ideas, well-written and unified framework. Therefore, the AC would like to recommend acceptance to this paper.